# Merging of ozone profiles from SCIAMACHY, OMPS and SAGE II observations to study stratospheric ozone changes

Carlo Arosio[1], Alexei Rozanov[1], Elizaveta Malinina[1], Mark Weber[1], and John P. Burrows[1]

[1]Institute of Environmental Physics, University of Bremen, Bremen

*Correspondence to:* carloarosio@iup.physik.uni-bremen.de

**Abstract.** This paper presents vertically and zonally resolved merged ozone time series from limb measurements of the SCanning Imaging Absorption spectroMeter for Atmospheric CHartographY (SCIAMACHY) and the Ozone Mapping and Profiler Suite (OMPS) Limb Profiler (LP). In addition, we present the merging of the latter two data sets with zonally averaged profiles from the Stratospheric Aerosol and Gas Experiment (SAGE) II. The retrieval of ozone profiles from SCIAMACHY and OMPS-LP is performed using an inversion algorithm developed at the University of Bremen. To optimize the merging of these two time series, we use data from the Microwave Limb Sounder (MLS) as a transfer function and we follow two approaches: (1) a conventional method involving the calculation of deseasonalized anomalies and (2) a 'plain-debiasing' approach, generally not considered in previous similar studies, which preserves the seasonal cycles of each instrument. We find a good correlation and no significant drifts between the merged and MLS time series. Using the merged data set from both approaches, we apply a multivariate regression analysis to study ozone changes in the 20–50 km range over the 2003–2018 period. Exploiting the dense horizontal sampling of the instruments, we investigate not only the zonally averaged field, but also the longitudinally resolved long-term ozone variations, finding an unexpected and large variability, especially at mid- and high-latitudes, with variations of up to 3–5 % per decade at altitudes around 40 km. Significant positive linear trends of about 2–4 % per decade were identified in the upper stratosphere between altitudes of 38 and 45 km at mid-latitudes. This is in agreement with the predicted recovery of upper stratospheric ozone, which is attributed to both the adoption of measures to limit the release of halogen-containing ozone-depleting substances (Montreal protocol) and the decrease in stratospheric temperature resulting from the increasing concentration of greenhouse gases. In the tropical stratosphere below 25 km negative but non-significant trends were found. We compare our results with previous studies and with short-term trends calculated over the SCIAMACHY period (2002–2012). While generally a good agreement is found, some discrepancies are seen in the tropical mid-stratosphere. Regarding the merging of SAGE II with SCIAMACHY and OMPS-LP, zonal mean anomalies are taken into consideration and ozone trends before and after 1997 are calculated. Negative trends above 30 km are found for the 1985–1997 period, with a peak of -6 % per decade at mid-latitudes, in agreement with previous studies. The increase of ozone concentration in the upper stratosphere is confirmed over the 1998–2018 period. Trends in the tropical stratosphere at 30–35 km show an interesting behavior: over the 1998–2018 period a negligible trend is found. However between 2004 and 2011 a negative long-term change is detected followed by a positive change between 2012 and 2018. We attribute this behavior to dynamical changes in the tropical middle stratosphere.

# 1 Introduction

The continuous monitoring of the stratospheric ozone layer is required to assess the impact of anthropogenic and natural processes (WMO, 2018). Variations of ozone concentration in time at different altitudes and latitudes respond to and are coupled with several dynamical and chemistry-related processes in the atmosphere.

Two important chemical forcings that have influenced globally the amount and distribution of stratospheric ozone over the last decades are the loadings of the so-called halogen-containing ozone-depleting substances (ODSs), that is halogen source gases released by human activities as chlorofluorocarbons (CFCs), and of greenhouse gases (GHGs) (WMO, 2018). The adoption of the Montreal Protocol and its amendments regulated the industrial production of chlorine and bromine compounds: in particular, the London amendment in 1990 called for a complete phase out of CFCs production by the year 2000, leading to
a decrease of their concentration in the stratosphere starting from the end of the 20th century (WMO, 2014). This decrease is expected to lead to a recovery of the ozone layer globally and in particular over the Antarctic region, which is affected by the spring-time ozone hole. On the other hand, the increasing concentration of GHGs such as $CO_2$ and $CH_4$ in the troposphere, is causing a cooling of the stratosphere, through radiative transfer feedbacks. This cooling leads to ozone increases due to the reactions R1 and R2:

$$O + O_2 + M \rightarrow O_3 + M \tag{R1}$$

$$O + O_3 \rightarrow O_2 + O_2 \tag{R2}$$

which have a strong temperature dependence (first predicted by Groves et al., 1978; Groves and Tuck, 1979). Cooling the stratosphere results in increased production and slower loss of ozone: a so called super recovery is thus expected (WMO,
2014). Models suggest that the combined effect of decreasing ODSs and increasing GHGs is going to lead to an increase in stratospheric ozone in the current and in the next decades. The magnitude of the recovery depends on the chosen scenario of anthropogenic emissions and on the actual decrease of ODSs (Waugh et al., 2009; Morgenstern et al., 2018).

    Another important species determining stratospheric ozone concentration belongs to the $NO_x$ family (NO, $NO_2$). The increasing tropospheric emissions of $N_2O$ or its longer residence time is causing a rise of NO concentration in the stratosphere
and a more efficient ozone destruction via the temperature-dependent $NO_x$ catalytic cycle. $N_2O$ is a long-lived GHG and it is expected to play a central role in the ozone recovery process over the next decades (Ravishankara et al., 2009). According to Portmann et al. (2012) it is rapidly becoming the most important ODS emitted by human activities. In addition, increasing emissions of $CH_4$ at the surface result in increasing $CH_4$ in the stratosphere and thus also of $HO_x$ (H, OH, $HO_2$). However the overall impact of increasing $CH_4$ is complex in the stratosphere: the ozone depletion by the $HO_x$ catalytic destruction cycles
occurs in the upper stratosphere, whereas the catalytic production of ozone is favoured by increasing $HO_x$ and sufficient $NO_x$ in the lower stratosphere.

    Changes in stratospheric dynamics also affect the latitudinal and altitudinal distributions of ozone. In particular, the speed of the tropical upwelling, i.e. the strength of the upward branch of the Brewer–Dobson circulation (BDC), is directly related to changes in the ozone distribution in the tropical lower and middle stratosphere. An acceleration of the stratospheric mean mass

transport has been predicted by several model studies (Garcia and Randel, 2008), but strong inter-annual variations prevent a significant recognition of this trend from observations. From monthly up to decadal time scales, ozone concentration is also influenced by many well known phenomena such as the 11 year solar activity cycle and solar proton events, the Quasi-Biennial Oscilation (QBO), El Niño Southern oscillation (ENSO), and volcanic eruptions (Tie and Brasseur, 1995; Soukharev and Hood, 2006; Randel et al., 2009; Park et al., 2017).

Interactions of all these chemistry- and dynamics-related contributions are therefore expected to result in a complex spatial pattern, depending on altitude, latitude and longitude. Therefore, to study long-term variations of the ozone distribution, there is a need for long-term consistent time series with a good temporal and spatial coverage of the whole globe.

Passive satellite instruments are able to provide good continuous global coverage and can be classified as nadir-viewing and limb-viewing (including occultation) sounders (Hassler et al., 2014). For stratospheric studies the limb geometry is the preferred choice, as it provides a relatively high vertical resolution. Several limb techniques have been developed over the last decades; in this paper we use data retrieved from measurements of limb scattering, limb emission and solar occultation instruments. A limb scattering sensor collects solar light scattered into the field of view of the instrument, whereas a limb emission instrument measures radiance emitted by atmospheric compounds in the infrared (IR) or microwave spectral region. Solar occultation sensors observe the solar disk and measure radiance attenuated along the ray-path through the atmosphere. The latter technique enables measurements of atmospheric trace gases profiles with a higher precision with respect to the other two but with a sparser spatial sampling, because the observations are only made at sunset and sunrise. The use of shortwave limb scatter technique was first successfully exploited by the NASA LORE/SOLSE (Limb Ozone Retrieval Experiment/Shuttle Ozone Limb Sounding Experiment) instrument launched in 1997 (McPeters et al., 2000). Two instruments soon followed: the Optical Spectrograph and Infrared Imager System (OSIRIS), launched in February 2001 (Llewellyn et al., 1997), and the SCanning Imaging Absorption spectroMeter for Atmospheric CHartographY (SCIAMACHY), launched in March 2002 (Burrows et al., 1995; Gottwald and Bovensmann, 2010). At the end of 2011, a few months before the end of the ENVISAT (Environmental Satellite) mission, the Ozone Mapping and Profiler Suite (OMPS) instrument was launched and it is still operational (Flynn et al., 2014). Stratospheric ozone profiles are currently monitored by limb sounders like the aging OSIRIS and the Microwave Limb Sounder on board the Aura satellite (Aura MLS, in the following referred to as MLS). In addition, solar occultation observations are currently done by the Canadian ACE-FTS (Atmospheric Chemistry Experiment - Fourier Transform Spectrometer) and MAESTRO (Measurement of Aerosol Extinction in the Stratosphere and Troposphere Retrieved by Occultation) instruments, launched in 2004 on board the SCISAT satellite, and the Stratospheric Aerosol and Gas Experiment (SAGE) III on the international space station, which was launched in 2017. The latter mission follows the successful SAGE II and SAGE III Meteor-3M instruments, which performed solar occultation observations from 1984 to 2005 and from 2002 to 2005, respectively.

In order to study the long-term changes in ozone vertical profiles and understand the impact of natural phenomena and anthropogenic activities on atmospheric ozone, single instrument time series are too short; several methodologies to consistently merge satellite data sets have been developed in the last years. In Harris et al. (2015), the authors considered several existing merged satellite data sets and examined separately the time spans before and after the peak in ODSs concentration at the end

of '90s. The authors combined trends from the different data sets and reported negative values in the upper stratosphere of -5 % to -10 % per decade before 1998, and a positive trend after 1998 of 2 % at mid-latitudes and 3 % in the tropics. Three different ways to compute uncertainties are also presented. They also stress different features visible in each single data set and the difficulty to establish the significance of trends in the latter period, requesting longer observational records, improvements

in the consistency of single data sets, and more accurate data merging with uncertainty estimates. Steinbrecht et al. (2017) updated this work, using several available merged satellite and ground-based data records, and computing an average ozone trend profile focusing on the 2000–2016 period. A significant increase of ozone in the upper stratosphere was reported, with values of 1.6–2.5 $\pm$1.1 % (1$\sigma$) per decade at mid-latitudes and 1.6 $\pm$0.6 % (1$\sigma$) % in the tropics. Sofieva et al. (2017) merged measurements from SAGE II with several other data sets homogenized within the Ozone-CCI (Climate Change Initiative)

project including OMPS limb observations. The authors used deseasonalized anomalies of zonal monthly mean time series to study trends over the 1980–2016 period. Before 1997 strong negative trends in the range from -4 to -8 $\pm$1.5 % (2$\sigma$) per decade were confirmed in the upper stratosphere. After 1997, the authors showed significant trends in the upper stratosphere at mid-latitudes reaching up to 2 $\pm$ 0.8 % (2$\sigma$) per decade in the northern hemisphere. Ball et al. (2018) applied a method independent from the ozone turnaround point, called dynamic linear method, to compute trends from several existing merged

data sets. The authors analyzed a longer period of time, together with improved merged time series and considered the lower stratospheric column instead of the ozone profile. With these adjustments, they showed for the first time some evidence of a negative trend in lower stratospheric ozone below 60° latitude. The authors claimed that the lower stratospheric decrease offsets the observed recovery in the upper stratosphere, leading to an overall decline of the stratospheric ozone column. This analysis has recently been challenged by Chipperfield et al. (2018), who showed that the apparent downward trend in the lower

stratosphere (ending in 2017) is a result of longer term variability in atmospheric dynamics. Bourassa et al. (2018) presented an updated trend analysis merging SAGE II with OSIRIS time series till 2017, after OSIRIS data were corrected for a drift in the tangent altitude registration of the instrument. The authors identified positive ozone trends post-1997 of about 1–3 % per decade above 25 km especially at mid-latitudes. In the lower stratosphere negative trends were found at all latitudes with significant values generally below 20 km.

Two other projects dealing with merging of satellite observations of several trace gas species are SWOOSH (Stratospheric Water and OzOne Satellite Homogenized) (Davis et al., 2016) and GOZCARDS (Global OZone Chemistry And Related trace gas Data records for the Stratosphere) (Froidevaux et al., 2015). The first study brought together satellite limb observations, providing several products such as water vapor and ozone mixing ratio profiles using different griddings on pressure levels starting from 1980. The second created time series of zonal monthly mean values of several trace gases using NASA satellites.

The LOTUS (Long-term Ozone Trends and Uncertainties in the Stratosphere) project is focused on investigating uncertainties in ozone trends, studying robust methods to merge data sets and homogenizing the trend evaluations (Petropavlovskikh et al., 2019).

This paper describes a merged ozone data set created using limb measurements from SCIAMACHY and OMPS. The two data sets were generated at the University of Bremen by applying a retrieval algorithm, which uses the same radiative transfer

model and spectroscopic databases and was individually optimized for SCIAMACHY and OMPS. The overarching scientific

objective was to derive consistent ozone data sets that could be merged with the help of a transfer function; the latter being necessary because of the limited overlap period of the two instruments (2.5 months). The merged data set comprises monthly averaged ozone profiles. One of the highlights of this merged data set, in comparison with those reported in several previous studies, except for SWOOSH, is that it is longitudinally resolved in steps of 5° latitude and 20° longitude. This enables us to

investigate long-term ozone changes as a function of altitude, latitude, and longitude over the past 15 years (2003 to 2018). In addition, we perform a merging of the two time series also in terms of ozone number density values, without subtracting the seasonal cycle from each data set. In order to investigate ozone trends over longer periods, we merged our new data sets with sparser ozone profiles retrieved from occultation measurements made by SAGE II. This SAGE-II/SCIAMACHY/OMPS merged data set is limited to zonal monthly mean anomalies. Section 2 of the paper describes the instruments, data sets, and

methods to retrieve ozone profiles used in this study. Section 3 introduces the merging of SCIAMACHY and OMPS limb data sets using two approaches. Section 4 reports about the long-term ozone changes, both zonally averaged and longitudinally resolved as derived from the SCIAMACHY/OMPS merged data. Results are discussed and compared with previous studies in Sect. 4. Section 5 introduces the merging of SCIAMACHY and OMPS zonal mean anomalies with SAGE II and discusses long-term ozone trends over the pre- and post-1997 periods.

## 2   Instruments and data sets

The SCIAMACHY instrument was launched in 2002 on board the ENVISAT satellite platform and made scientific measurements from August 2002 until April 2012, when a failure in the platform-to-ground communication occurred. In the limb mode, SCIAMACHY observed the atmosphere in flight direction and scanned horizontally, covering 960 km across-track in four steps, and vertically every 3.3 km. The instrument had a wide spectral coverage, collecting radiances in 8 channels span-

ning from 240 to 2380 nm, with a spectral resolution varying from 0.22 to 1.48 nm depending on the channel (for a detailed description of the instrument see Burrows et al., 1995; Gottwald and Bovensmann, 2010).

The OMPS instrument was launched at the end of 2011 on board the Suomi-NPP satellite platform (Flynn et al., 2014). The suite is composed of three instruments, only data from the Limb Profiler (LP) used for this study (in the following referred to as OMPS-LP). The instrument looks backwards with respect to the flight velocity vector. It observes the whole atmospheric

range simultaneously without scanning, via three vertical slits. The central slit is aligned with the satellite ground track and the other two are sideways, so that the instrument performs measurements at three viewing angles, which differ horizontally by 4.25°. The sensor collects spectral radiance on a two-dimensional charged-coupled device (CCD) through two apertures and at two integration times, to account for the wide dynamic range of the scattered radiance. The CCD pixels are then sampled to obtain a single picture of the atmospheric state and interpolated to derive level 1 gridded data (L1G). OMPS-LP has a spectral

coverage from 280 to 1000 nm with a spectral resolution increasing from 1 nm in the ultraviolet (UV) region to 30 nm in the near-IR.

In Table 1 some details of the SCIAMACHY, OMPS-LP, MLS and SAGE II instruments are reported.

**Table 1.** Main characteristics of SCIAMACHY, OMPS-LP, MLS and SAGE II instruments.

|  | SCIAMACHY | OMPS-LP | MLS | SAGE II |
|---|---|---|---|---|
| Data time series | 01.2003–03.2012* | 03.2012–06.2018* | 01.2005–12.2016* | 01.1985–08.2005* |
| Spectral coverage | 240–2380 nm | 280–1000 nm | 118 GHz – 2.5 THz | 385–1020 nm |
| Spectral resolution | 0.22–1.48 nm | 1–30 nm | ** | 1–2 nm |
| Instantaneous field of view [km] | 2.6 | 1.5 | 1.5–3 | 0.5 |
| Number of observations per orbit | ∼120 | 180 (each slit) | ∼120 (day-side) | 2 |
| Latitude coverage | 83.5° S–83.5° N | 81.3° S–81.3° N | 81.8° S–81.8° N | 80.0° S–80.0° N |
| Equatorial crossing time | 10:00 | 13:30 | 13:45 | - |
| Level 2 data version | 3.5 | 2.6 | 4.2 | 7.0 |

\* used in this paper
\*\* see details in Waters et al. (2006)

In this study we use version 3.5 of SCIAMACHY ozone profile retrieval and OMPS-LP version 2.6: both products were created at the University of Bremen using the SCIATRAN software package (v3 for SCIAMACHY and v4 for OMPS-LP) which includes a radiative transfer model and a retrieval algorithm (Rozanov et al., 2014). In particular, v8 L1 SCIAMACHY and v2.5 L1 OMPS-LP data were processed. As discussed above and listed in Table 1, differences in terms of spectral coverage and resolution, observation method and radiance collection prevented a direct application of SCIAMACHY's retrieval scheme to OMPS-LP. However for the retrieval of both data sets we used the same spectroscopic databases (Serdyuchenko et al., 2014; Bogumil et al., 2000) and the same initialization for atmospheric composition and optical parameters. Both algorithms are based on a Tikhonov regularization scheme and use spectral windows in the UV Hartley-Huggins and in the visible Chappuis ozone bands. The SCIAMACHY ozone profile retrieval algorithm exploits the sun-normalized limb radiance measurements for Huggins and Chappuis bands, while measurements in the Hartley band are normalized to an upper-altitude tangent height. For OMPS-LP, measurements of the solar spectral irradiance are not directly reported in v2.5 L1G data, so we normalize the radiance in all absorption bands using upper-altitude tangent heights. In both cases we also take into account the absorption of $NO_2$ and $O_4$, using the same cross sections but convolved to the respective resolution of the instruments. The weighting functions of the surface reflectance are included in the fit procedure. The presence of a cloud in the instrument field of view is detected following the color index approach (Eichmann et al., 2016). Aerosol extinction profiles are retrieved for OMPS-LP using the methodology described in Rieger et al. (2018), whereas for SCIAMACHY climatological profiles are used. SCIAMACHY profiles are reported from 8 to 64 km with a vertical sampling of 3.3 km and a vertical resolution of 2.6 km, OMPS-LP profiles span from 12 to 60 km with typical vertical resolution of 3 km and a sampling every 1 km. Only measurements from the central slit of the OMPS-LP instrument are used in this study; data from the lateral slits are planned to be used when the issues related to the tangent altitude registration of the instrument, currently under investigation by NASA (Moy et al., 2017), are solved.

For more details about the University of Bremen OMPS-LP retrieval algorithm implementation and validation readers are referred to Arosio et al. (2018); for a description of SCIAMACHY retrieval and the validation of the ozone profiles to Jia

et al. (2015). Briefly, in Arosio et al. (2018) it has been shown that the retrieved OMPS-LP profiles averaged on a yearly basis agree with MLS within 5–10 % between 20 and 50 km, while below 20 km discrepancies are larger especially in the tropical upper troposphere and lower stratosphere. Also the validation with ozonesondes showed an agreement within ±7 % between 20 and 30 km in five chosen latitude bands, with a larger overestimation of the retrieved profiles in the tropics below 22 km.

The validation of SCIAMACHY v3.5 against single ozonesondes stations performed in Jia et al. (2015) showed an agreement between the two data sets within 10 % between 20 and 30 km, with discrepancies in the tropics above 22 km generally below 5 %.

The first study which addressed a possible drift of SCIAMACHY v3.5 with respect to other ozone satellite data sets, is Sofieva et al. (2017). The authors stated that evaluating and inter-comparing the anomalies of the considered instruments,

among which SCIAMACHY starting from August 2003, they did not find statistically significant drifts with respect to the median anomaly. Kramarova et al. (2018) reported an estimation of v2.5 NASA OMPS-LP ozone profiles drift with respect to MLS, finding positive values up to 0.5–1.0 % $yr^{-1}$ above 35 km.

The MLS instrument was launched on board the Aura satellite and started atmospheric observations in July 2004, observing the thermal emission from atmospheric trace gases in the millimeter/sub-millimeter spectral range. It scans the Earth limb 240

times per orbit providing retrievals of day- and nighttime profiles of several gases including ozone. For a detailed description of the MLS instrument readers are referred to Waters et al. (2006). In this paper, the version 4.2 of MLS level 2 (L2) data is used as a transfer function in the SCIAMACHY/OMPS-LP merging procedure. Quality flags and recommendations reported in Livesey et al. (2017) are used in the study. Hubert et al. (2016) investigated the stability of MLS ozone data set and found no significant drifts over the entire stratosphere.

SAGE II was launched in October 1984 on board the Earth Radiation Budget Satellite (ERBS) and operated until August 2005. The instrument had a sunphotometer collecting solar radiance attenuated by the atmosphere in seven wavelength ranges using the occultation technique. Due to the occultation viewing geometry, the observations of SAGE II are sparse in comparison to that from limb instruments. It performed measurements only twice per orbit, resulting in 30 observations per day. The occultation geometry, however, yields a higher signal to noise ratio and the ozone profiles are provided with a vertical resolution

of 0.5 km from cloud top to 60 km. For a more detailed overview of the instrument, readers are referred to McCormick (1987). In this study we use version 7.0 of SAGE II L2 data (Damadeo et al., 2013).

## 3 Merging the data sets

When merging different data sets, calibration discrepancies between the instruments as well as eventual drifts and jumps in the time series must be accounted for (Hubert et al., 2016). As the overlap period of SCIAMACHY and OMPS missions is only

about 2.5 months, i.e. too short for a reliable bias correction, we select a reference satellite data set to be used as an external transfer function. For this purpose, MLS was chosen because of the stability and reliability of its measurements, the extensive overlapping period with both instruments, its broad latitude coverage, and its dense sampling. In particular, we use daytime MLS data from January 2005 until December 2016. For each day, we take only MLS measurements which are made within

the latitude range covered by OMPS-LP and SCIAMACHY. The presence of the so-called South Atlantic Anomaly (SAA) is filtered using for MLS and OMPS-LP the SAA flag provided in their respective L2 data and applying for SCIAMACHY a rectangular exclusion mask over the [-70°, -20°] latitude and [270°, 360°] longitude range. The SCIAMACHY data set is considered in this study starting from January 2003 till March 2012. April 2018 is excluded because data for the first 8 days only are available, whereas 2002 data are excluded because of the large discrepancies of SCIAMACHY anomalies with respect to other satellites identified by Sofieva et al. (2017). OMPS-LP data from March 2012 until June 2018 are used for merging. All profiles are provided in units of ozone number density on a geometric altitude grid. Volume mixing ratio (VMR) ozone profiles from MLS on a pressure grid are converted to geometric altitude vs. number density using collocated pressure information from the ECMWF (European Centre for Medium-Range Weather Forecasts) ERA-Interim database and temperature profiles retrieved by MLS. In Appendix A we show the sensitivity of the MLS average ozone distribution and of the computed ozone trends to a change of the reanalysis database. In particular, we use data from ECMWF ERA-Interim and MERRA-2 (Modern-Era Retrospective analysis for Research and Applications, version 2, Gelaro et al. (2017)). The effect on ozone trends of using different reanalysis databases providing pressure information is negligible, within -0.25 and +0.5 % at most of the altitudes.

Different ways to bin the satellite data have been studied in order to find an optimal tradeoff between sufficiently high spatial and temporal resolution of the merged product and the number of measurements in each bin, for the values to be representative. Two optimal sets of values are identified: a longitudinally resolved product, with monthly mean values on a 5° latitude and 20° longitude grid and a zonally averaged product with a temporal resolution of 10 days and a latitude resolution of 2.5°. In both cases we find on average 50–100 profiles in each bin. The vertical grid used for the merged profiles has evenly spaced steps of 3.3 km, which corresponds to the typical SCIAMACHY vertical sampling. MLS and OMPS-LP profiles with denser vertical sampling are linearly interpolated to this common grid.

In this paper we consider the longitudinally resolved ozone profile product, i.e. monthly averaged profiles every 5° latitude and 20° longitude. In some cases however we don't show the longitudinally resolved results, either for lack of space or because the zonal averages are directly comparable with previous studies. In this case, the average over longitudes is performed on the level 3 data prior to any further computations (e.g., trends, differences, etc.). Figure 1 shows the number of measurements available for SCIAMACHY and OMPS-LP in each altitude and latitude bin as a function of time. These values have to be divided by 18, the number of longitudinal bins, to determine the number of measurements that contribute to each longitudinally resolved monthly mean value. The density of measurements increases in 2012, because OMPS-LP has a higher sampling per orbit than SCIAMACHY, as reported in Table 1.

Two approaches are used to merge the SCIAMACHY and OMPS-LP data. In the first one, the so-called 'plain-debiasing' approach, the seasonal cycle (SC) of each instrument is kept: one data set is shifted with respect to the other, with the help of the transfer function, to remove the offset between the two. In the second one, the so-called 'anomalies' approach, which is similar to that used by Sofieva et al. (2017), the SC of each single instrument data set is determined and anomalies are calculated independently for each data set. Then the offset between SCIAMACHY and OMPS-LP is subtracted using the MLS anomalies as a transfer function. We study the SCs of the three instruments to asses how well they agree and whether they need to be

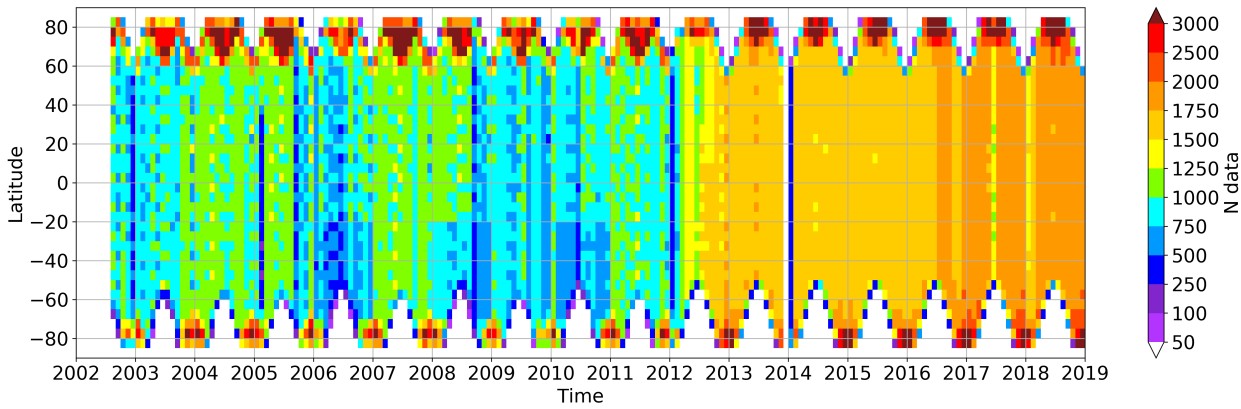

**Figure 1.** Number of SCIAMACHY and OMPS-LP observations as a function of time and latitude in each 5° zonal monthly bin.

subtracted before merging. Figure 2 shows the SCs of the three ozone profile data records in number density [molec $cm^{-3}$] at different altitudes and latitudes.

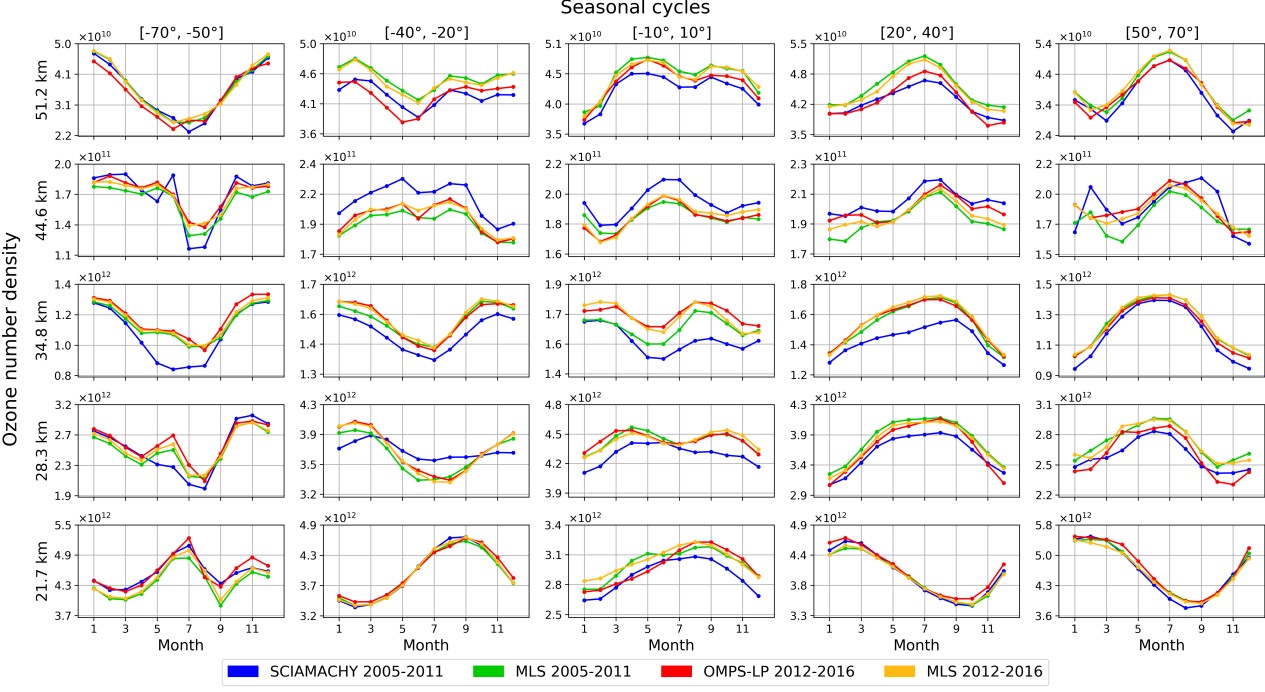

**Figure 2.** Seasonal cycle (SC) for the three instruments as a function of latitude and altitude, in terms of ozone number density [molec $cm^{-3}$]. MLS SC is plotted for the overlapping period with SCIAMACHY (2005–2011) and with OMPS-LP (2012–2016).

The SCIAMACHY ozone SC is compared to that from MLS profiles, computing it for both instruments over the period 2005–2011, whereas the OMPS-LP ozone SC is compared to that from MLS profiles for the period 2012–2016. At first glance, there is generally good agreement; however, discrepancies are visible in terms of additive bias, multiplicative bias (different amplitude of SC) and shape of the SC between the instruments. Through the merging process additive biases are subtracted via the 'plain-debiasing' procedure, whereas the multiplicative bias and the discrepancies related to the different shape of the SC are accounted for when calculating anomalies. Two clear examples for these types of discrepancies are seen in the latitude band [-40°, -20°] at two altitudes (see Fig. 2):

1. at 34.8 km the SCs of the three instruments show the same shape but different absolute values;

2. at 28.3 km SCIAMACHY SC has a significantly smaller amplitude with respect to the MLS and OMPS-LP.

Differences in the amplitudes are caused by the different vertical sampling of the instruments and by the interpolation procedure we adopted; they are more pronounced at latitudes and altitudes where the transition between semi-annual to annual cycle occurs. In addition, the natural variability of the atmosphere plays an important role, with the SC that naturally evolves with time: we notice, for example, that the SC measured by MLS varies between the two considered periods, in particular at 34.8 km in the tropics, where a change of up to 5–7 % occurs.

As SCIAMACHY and OMPS-LP have a very similar geometry of observation, a comparable latitude coverage and their SCs do not differ significantly except for few latitudes and altitudes, the first approach for merging the two time series consists in a 'plain debiasing' of the data sets with respect to MLS. The bias is defined for each latitude, longitude and altitude as follows:

$$BIAS_{SCIAMACHY}(\phi,\theta,z) = mean(SCIAMACHY_{2005-2012}(\phi,\theta,z)) - mean(MLS_{2005-2012}(\phi,\theta,z)) \tag{1}$$

$$BIAS_{OMPS}(\phi,\theta,z) = mean(OMPS_{2012-2016}(\phi,\theta,z)) - mean(MLS_{2012-2016}(\phi,\theta,z))$$

In these and following equations, ozone profiles from each instrument are considered as binned monthly averages, interpolated to a common altitude grid. These biases are then applied to the OMPS-LP time series in such a way to conventionally keep the SCIAMACHY mean level as absolute reference as follows:

$$OMPS_{deb}(\phi,\theta,z) = OMPS(\phi,\theta,z) - BIAS_{OMPS}(\phi,\theta,z) + BIAS_{SCIA}(\phi,\theta,z) \tag{2}$$

In this way, any offset between SCIAMACHY and OMPS-LP is accounted for with the help of MLS as a transfer standard. The merging is then achieved by concatenating the two data sets, in terms of ozone number density, and computing average values from SCIAMACHY and OMPS-LP over the two months of overlap, i.e. February–March 2012. We exclude all bins where the number of observations is lower than 10 or where the measurements from one of the instruments are not available. Figure 3 shows relative differences between the merged data set and MLS time series (after the subtraction of its bias with respect to SCIAMACHY) as a function of latitude for several altitudes.

Relative differences for the 'plain-debiased' merged time series are computed as follows:

$$Rel\ Diff(\phi,\theta,z) = (Merged(\phi,\theta,z) - MLS(\phi,\theta,z))/(Merged(\phi,\theta,z) + MLS(\phi,\theta,z)) * 200 \tag{3}$$

Differences are within ±10 % between 20 and 50 km and between 50° S and 50° N. Dashed vertical lines indicate the transitions between the two instruments. Over the SCIAMACHY measurement period, a small SC signature is observed, especially at 30–35 km at mid-latitudes and at 40–45 km at higher latitudes; these differences are already visible in Fig. 2. In the second half of the time series, less pronounced SC signatures are seen, particularly between 35 and 45 km. Below 20 km

the differences increase rapidly showing strong seasonal pattern. Above 50 km, we notice a variation of the relative differences with time, suggesting the presence of drifts with respect to MLS within the time span of each instrument. Caution is therefore required in interpreting the computed trends above 50 km. At these altitudes diurnal variation of ozone have to be accounted for, as showed by Sakazaki et al. (2013). This was not done in our study, because the equatorial crossing time of the two instruments is around noon and differs by only 3.5 h: this would lead to a systematic discrepancy in ozone that we estimate

to be about 1–2 % at 30–40 km. Furthermore, the expected systematic bias between the two instruments is largely removed by the debiasing procedure, even though not completely, because variations with time of this systematic discrepancy may not be accounted for by a 'plain-debiasing'. In addition, a technical change in the L1 processing of OMPS-LP UV data at the beginning of 2014 affects the OMPS-LP UV retrieval and leads to a jump above 50 km between the 2012–2013 period and the last three years of observations. Towards the polar regions, we notice increasing relative differences with respect to MLS,

particularly above 40 km and below 25 km. In summary, we recommend the use of the 'plain-debiased' time series within ±60° latitudes and the 20–50 km altitude range.

The second approach to merge data follows that from Sofieva et al. (2017) and comprises computing the deseasonalized relative anomalies from each data set and then debiasing them using MLS data. This is a common procedure when merging several data records, in order to account for the different geometry and atmospheric sampling by each sensor. The SC for each

month of the year, $m$, and the (relative) anomalies, $\Delta$, are defined as:

$$SC_m = \frac{1}{N_m} \sum_{j=1}^{N_m} O_3(t_j) \tag{4}$$

$$\Delta(t_m) = \frac{O_3(t_m) - SC_m}{SC_m} \tag{5}$$

$$\tag{6}$$

for SCIAMACHY, OMPS-LP and MLS, where $N_m$ is the number of available monthly mean values $O_3(t_j)$ for the month

of the year $m$ in each time series. The SC is computed for each instrument considering their complete time series. Then, the anomalies $\Delta(t_m)$ of SCIAMACHY and OMPS-LP are debiased using MLS anomalies as a transfer function as described by Eqs.(1) and (2). The merging is performed in the same way as done for the first approach. Figure 4 shows the time series of absolute differences between the merged anomalies and MLS anomalies as a function of latitude for several altitudes, in percentage, computed as follows:

$$Diff(\phi, \theta, z) = (Merged(\phi, \theta, z) - MLS(\phi, \theta, z)) * 100 \tag{7}$$

The differences are generally within ±5 % also towards the polar regions between 20 and 50 km for both SCIAMACHY and OMPS-LP periods, showing a smaller magnitude and a better consistency over the whole time series with respect to Fig. 3.

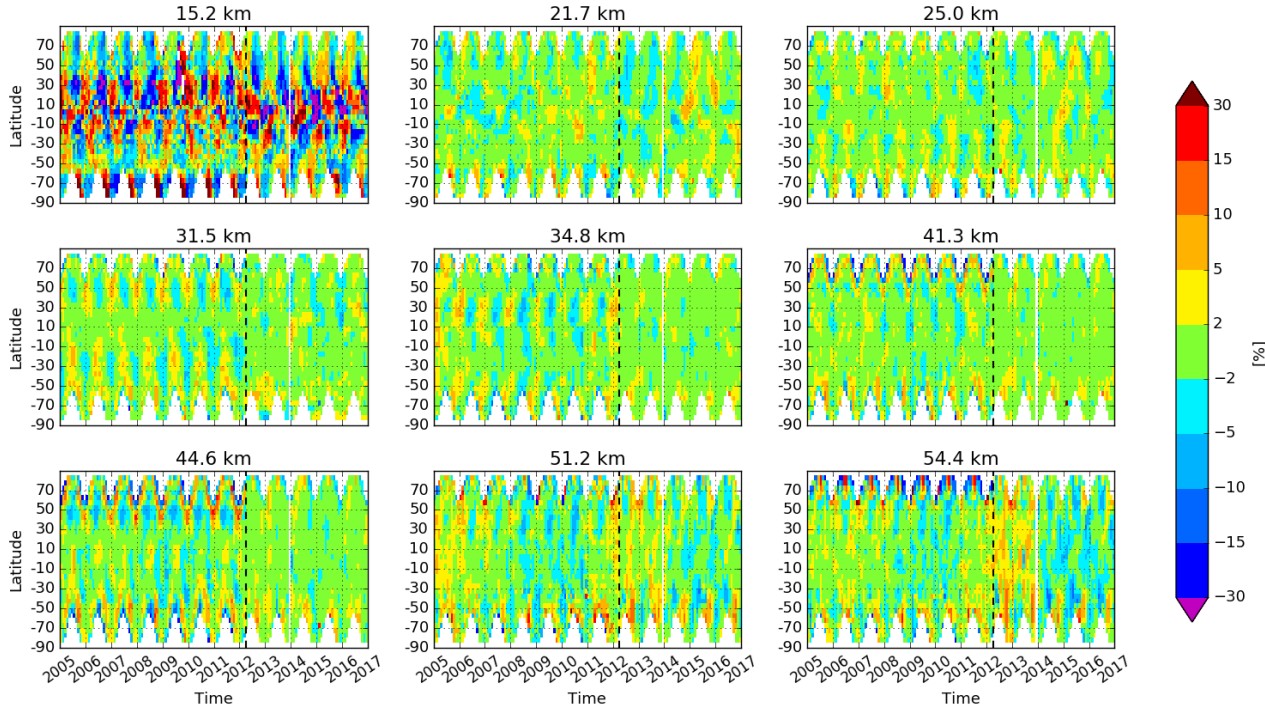

**Figure 3.** Relative differences of the debiased merged time series ('plain-debiasing' approach) with respect to MLS as a function of latitude for several altitudes, computed according to Eq. (3). The vertical dashed lines indicate the transition between SCIAMACHY and OMPS-LP data sets. MLS data has been offset to SCIAMACHY before the comparison.

Above 50 km, the presence of a drift within the single data sets is again observed, whereas the jump observed in Fig. 3 between the first two years of OMPS-LP lifetime and the rest of the time series is strongly reduced. Below 20 km the pattern becomes rather chaotic in this case as well, also due to the fact that low values of ozone number density, especially in the tropics, amplify the relative differences. We recommend the use of this data product within $\pm70°$ latitudes and over the 20–50 km altitude range.

To check the consistency of the SCIAMACHY/OMPS-LP merged data set with respect to MLS, we compute the correlation coefficient and the drift for each latitude-altitude bin with respect to MLS over the period 2005–2016. The drift is computed as the linear change of the differences (either relative Eq. (3) or absolute Eq. (7) for the 'plain-debiased' data set and anomalies respectively) between the merged time series and MLS data, accounting also for seasonal variations as a sum of harmonic terms with periods of 6 and 12 months in the fit. Figure 5 shows in panel (a) the Pearson correlation coefficient as a function of altitude and latitude for the zonally averaged merged data set with respect to MLS, for the 'plain-debiased' merged data set (first approach). The correlation coefficient is high being typically above 0.8 between 20 and 50 km and within $\pm70°$ latitudes. A very similar result is obtained for the deseasonalized anomalies (see the Supplements, Fig. S1). Pearson correlation coefficient

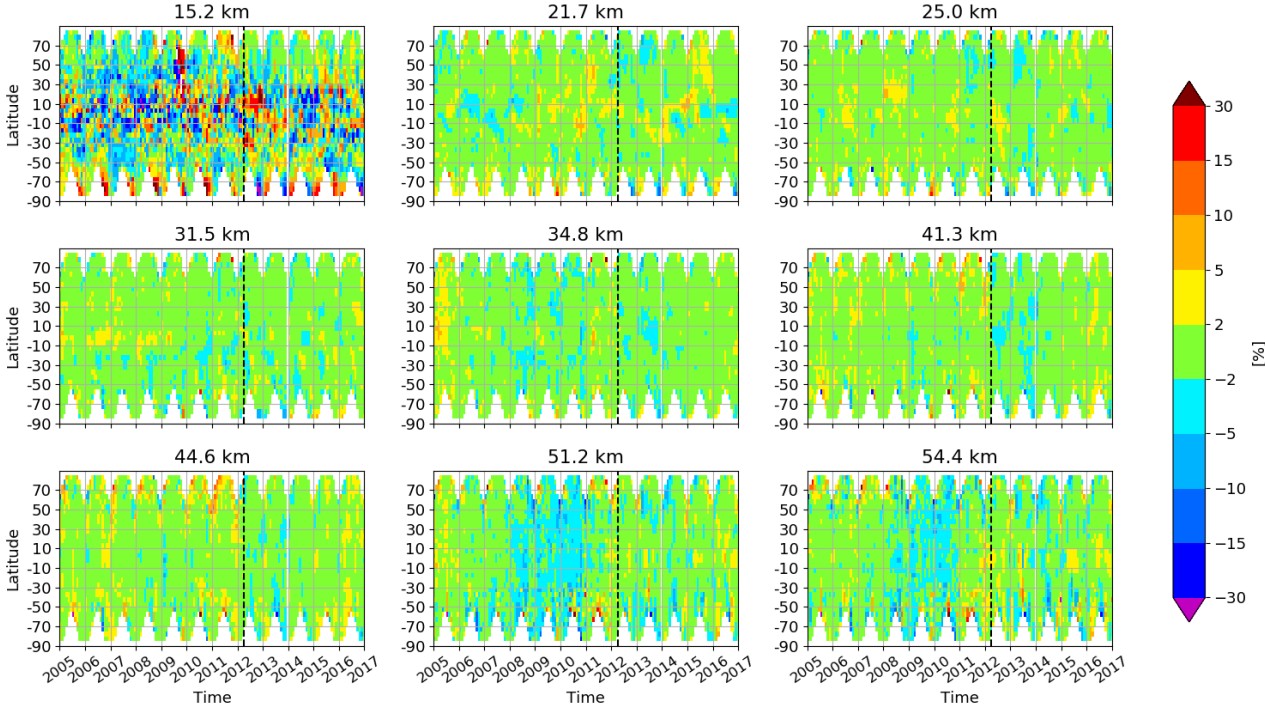

**Figure 4.** Differences of the merged relative anomaly time series with respect to MLS anomalies as a function of latitude at selected altitudes, computed according to Eq. (7). The vertical dashed lines indicate the transition between SCIAMACHY and OMPS-LP data sets. MLS data has been offset to SCIAMACHY before the comparison.

values are in that case slightly lower because the strong SC removed in the anomalies contributes largely to the correlation. Panel (b) of Fig. 5 shows the drift of the merged data set with respect to MLS, in terms of % per decade; dashed areas in this and the following figures indicate non-significant values, using a 95 % confidence level. The drift is positive only in the tropical lower stratosphere and above 40 km towards the polar regions but values are generally non-significant between 20 and 50 km:

5   this means that the three debiased data sets (MLS and debiased SCIAMACHY and OMPS-LP) are consistent with each other over the 11 years of comparison and the long-term ozone changes from the merged data set can be computed with high degree of confidence. Very similar results for the drift are obtained using anomalies time series, whose respective plot can be found in the Supplements (Fig. S1). A plot of the longitude-resolved drift values is also shown in the Supplements, Fig. S2: we notice in this plot a longitudinal structure: even though the drift is mostly non-significant, negative values are found in the $[0°, 80°]$

10   longitude band, whereas positive values are detected within $[100°, 260°]$ longitude and close to zero values elsewhere.

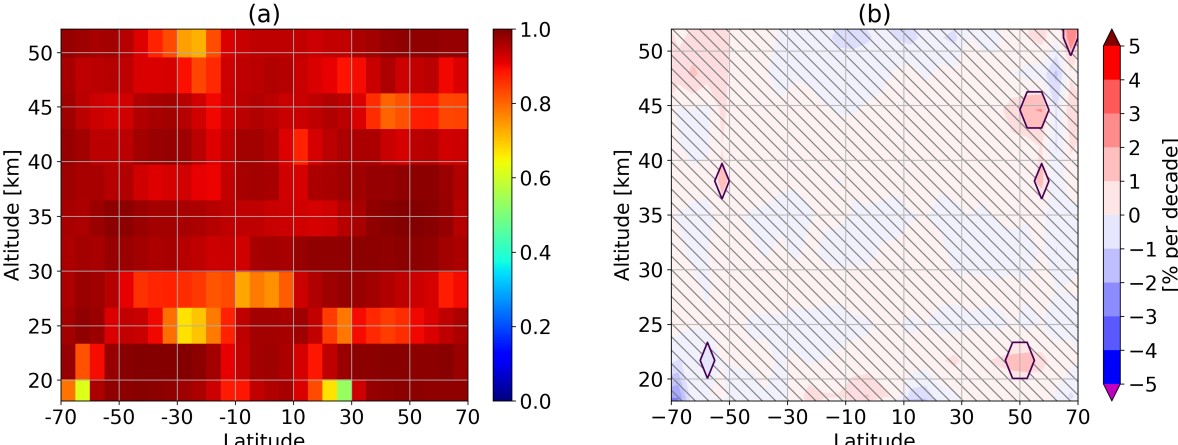

**Figure 5.** Panel (a): Pearson correlation coefficient of the merged debiased data set with respect to MLS time series over 2005–2016. Panel (b): drift of the merged debiased time series with respect to MLS in % per decade, (differences computed according to Eq. (3)); dashed areas identify regions where the drift is not statistically significant.

## 4 Trend analysis

### 4.1 Multivariate linear regression terms

To study recent long-term ozone variations with the new merged data sets, we have selected the period January 2003–June 2018, consisting of 186 months. We follow a standard approach, applying an unweighted multilinear regression (MLR) model, accounting for several factors affecting ozone variability in the stratosphere. The weighting of each value by using the reciprocal of its corresponding squared standard deviation, i.e. $\sigma^2(t_m)$ in Eq. (6), has been tested but does not affect significantly the results. The autocorrelation of the data set with one month lag is accounted for, assuming the noise, $N$, to be an autoregressive process of the first order (Weatherhead et al., 1998). The following terms are considered in the MLR (Gebhardt et al., 2014):

$$O_3(t) = c_0 + c_1 t + \sum_{j=1}^{2}\left(c_{2j}sin(\frac{2\pi j t}{12}) + c_{3j}cos(\frac{2\pi j t}{12})\right) + QBO(t) + Solar(t) + ENSO(t) + N \tag{8}$$

*or*

$$\boldsymbol{O_3} = \mathbf{X}\boldsymbol{\beta} + \boldsymbol{N}$$

where t is the time in months, $c_i$ are the regression coefficients, contained in the $\beta$ vector and the t-th row of the $\mathbf{X}$ matrix contains the values of the fit terms for the selected t. The ozone time series can be either in terms of number density [molec cm$^{-3}$] or relative anomalies (multiplied by 100) [%]. The trend uncertainty and thus the significance of the linear trend values are computed from the covariance matrix of the regression coefficients; the trend is significant at the 95 % significance level if the

following condition is fulfilled:

$$\left| \frac{c_1}{\sigma_{c_1}} \right| >= 2 \tag{9}$$

All trends shown here are expressed in % per decade: the 'plain-debiased' time series are regressed in terms of [molec cm$^{-3}$] and the obtained trend values are divided by the averaged ozone series in each bin.

The linear term determined from Eq. (8) is the ozone trend at a given altitude, latitude and longitude. The harmonic terms with a period of 6 and 12 months are considered only for the 'plain-debiased' merged data set to approximate the seasonal behavior. For the 50–60° N latitude band, the seasonal variability of ozone below 25 km is approximated by using a term containing the eddy heat flux time series instead of harmonic terms. The eddy heat flux is used as a proxy for the strength of the BDC (Weber et al., 2011). Indeed in this latitude band, the strong inter-annual variability related to the wave forcing might

be insufficiently modeled when using harmonic terms only. As a consequence, the two months lagged eddy heat flux at 50 hPa from ERA-Interim is integrated over each year starting from October and used as a fit proxy (Gebhardt et al., 2014).

    The QBO is a quasi-periodic variation of the tropical wind direction in the tropical stratosphere: easterly and westerly wind regimes propagate downward with a variable period of approximately 28 months at a given altitude level. Even though it is a tropical phenomenon, the effects of this variable wind pattern on ozone are not confined to the tropical region: they extend to

mid- and high-latitudes and are associated with the secondary meridional circulation (Baldwin et al., 2001). Park et al. (2017) illustrated the effects of the QBO on ozone profiles as a function of altitude with two peaks in ozone changes found at 20–27 km and at 30–38 km, showing opposite phases in the tropics and being in phase at mid-latitudes. In this study the influence of QBO is accounted for by considering the monthly average of the zonal wind components measured at 10 and 30 hPa by sondes launched at Singapore station (available at http://www.geo.fu-berlin.de/en/met/ag/strat/produkte/qbo/index.html) as a

fit proxy. This combination of tropical zonal winds is used at all altitudes and latitudes as follows:

$$QBO(t) = c_{4a}QBO_{10_{hPa}}(t) + c_{4b}QBO_{30_{hPa}}(t) \tag{10}$$

    The solar activity has a noticeable impact on ozone especially in the upper stratosphere as a consequence of e.g. the 11 year cycle and associated strong solar proton events. Several studies based on satellite data sets showed the presence of an in-phase solar cycle. Soukharev and Hood (2006) studied a 25 year period and found statistically significant ozone variation between

the maximum of the solar cycle and its minimum in the upper and in the lower stratosphere. The main contribution to the total ozone column response to the 11 year solar cycle is found to come mainly from altitudes below 25 km. The correlation is found to be positive and without time lag. More recently Maycock et al. (2016) compared the solar-ozone response from several recently updated satellite time series. In particular, they used the updated v7.0 of SAGE II data, finding reduced variations in ozone in the tropical upper stratosphere (∼1 %) due to the solar cycle. This is in agreement with their analysis of Solar

Backscatter Ultraviolet Instrument (SBUV) v8.6 data set. As a proxy for the solar activity we use Mg II index, which is the core-to-wing ratio derived from the Mg II doublet that is known to be highly correlated to solar irradiance variability from the UV to the extreme-UV (Snow et al., 2014). The composite Mg II data set we use was derived at the University of Bremen from the Global Ozone Monitoring Experiment (GOME), SCIAMACHY, GOME-2A and GOME-2B data (and available at

http://www.iup.uni-bremen.de/UVSAT/Datasets/mgii). The solar proxy applied to all latitudes and altitudes is given by:

$$Solar(t) = c_5 MgII(t) \tag{11}$$

A further dynamical process impacting stratospheric ozone is the ENSO. This ocean-atmosphere coupled oscillation over the tropical eastern Pacific Ocean has been shown to impact the BDC and is responsible for temperature anomalies in the upper troposphere and lower stratosphere, leading to longitudinally dependent modifications of ozone in this region (Randel et al., 2009). We include the El Nino 3.4 index as a fit proxy for ozone variations in the lower stratosphere, which is based on sea surface temperature anomalies averaged from 5° S–5° N and 170°–120° W. The data time series is available at $http://www.esrl.noaa.gov/psd/gcos\_wgsp/Timeseries/Nino34/$. In particular, we considered a proxy based on a combination of El Nino 3.4 index anomalies and its derivative, in order to account for the time lag between the ENSO proxy and its signature in the ozone time series, as follows:

$$ENSO(t) = c_6[N_{34} + \frac{dN_{34}}{dt}\Delta(t)] \tag{12}$$

where $\Delta$ indicates the time lag in months. An iterative procedure is used to assess $\Delta$. Starting from a time lag of two months the MLR is repeated, updating the time lag at each iteration until it approaches a fraction of a month. The final time lag is allowed to vary between 0 and 12 months. If it does not converge within 10 iterations or exceeds this range, the ENSO proxy is not used in the regression. ENSO is taken into consideration only in the tropical regions (20° S–20° N) below 25 km.

## 4.2  Zonally and longitudinally resolved long-term ozone variations

Figure 6 shows long-term ozone changes of zonally averaged ozone as a function of latitude and altitude calculated using the MLR model applied to the two versions of SCIAMACHY/OMPS-LP merged data sets: in panel (a) considering the anomalies data set and in panel (b) following the 'plain-debiasing' approach. The longitudinally resolved trends are reported in the Supplements, Fig. S3. The general picture in the two panels is similar, noting that trend values in panel (b) are slightly larger compared to those in panel (a). This fact may be related to the method use to compute trends: in the anomalies strategy, the absolute anomalies are divided by the SC and then directly used to compute trends in % per decade. In the 'plain-debiasing' approach, trends are computed using the time series in number density and then normalized to the average ozone values at each altitude, latitude and longitude, to obtain values in terms of % per decade. Long-term changes are only statistically significant at mid-latitudes in the upper stratosphere. In this region the long term change is about 3–4 % per decade. This increase shows an asymmetry between the two hemispheres, with higher values at northern high-latitudes, also seen in other studies such as Bourassa et al. (2018). As discussed in Sect. 1, a recovery of upper stratospheric ozone as a consequence of decreasing ODSs and increasing GHGs emissions is expected and in agreement with recent studies (e.g WMO, 2018; Petropavlovskikh et al., 2019). This is because at these altitudes the production of ozone results from the photolysis of ground state molecular oxygen, $O_2(^3\sum_g^-)$ and the subsequent three body reaction of ground state oxygen atoms, $O(^3P)$, with $O_2(^3\sum_g^-)$ whereas ozone is lost by temperature-dependent catalytic odd oxygen cycles involving $ClO_x$, $BrO_x$, $HO_x$ and $NO_x$. Above 48 km in the tropics the negative trends appear significant. As discussed in Sect. 3, these values have to be taken with caution. In addition, we tested

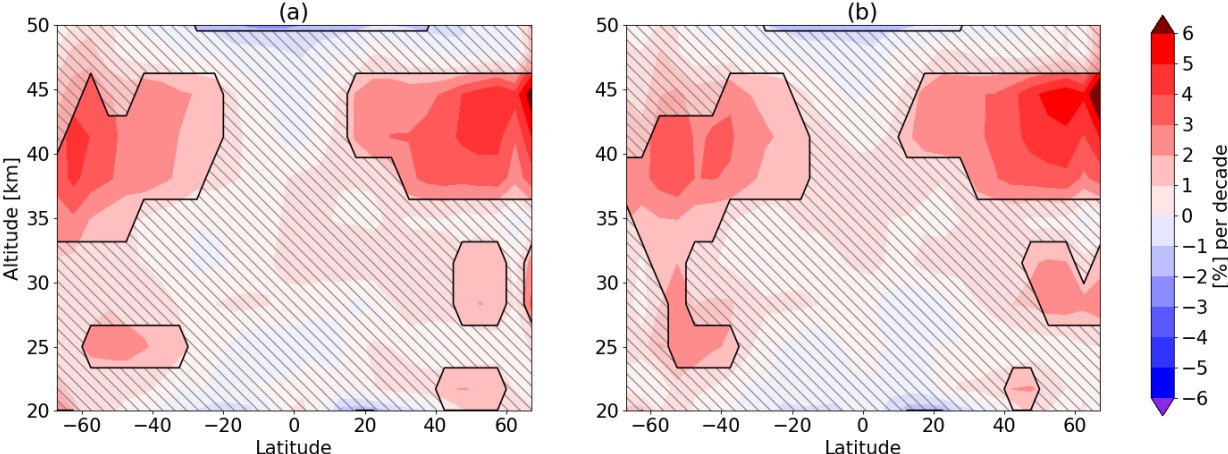

**Figure 6.** Zonal mean linear long-term ozone changes over the 2003–2018 period derived from the SCIAMACHY/OMPS-LP merged data sets: panel (a) shows the results using anomalies, panel (b) shows the results the 'plain-debiased' data set. Dashed areas indicate non-significant trends.

the robustness of the trends by changing the starting point of the time series. When the time series starts from mid-2003 or beginning of 2004, the negative trend between 45 and 50 km is strongly reduced and is not significant anymore, whereas the positive trends get larger at mid-latitudes (of about 1 %). This implies that an unresolved instrumental issue at the beginning of the SCIAMACHY data set at these altitudes exists or that the starting point of the time series is not optimal to draw conclusions

5    on long-term trends. In comparison with Ball et al. (2018), we do not identify extensively negative trends below 25 km in the tropics: at these altitudes in the [-30°, 30°] latitude range negative trends are observed but are generally not significant. As can be seen in the Supplements, Fig. S3, only around 18–20 km in some longitude bins a statistically significant ozone decrease is detected. We also performed the merging procedure and the trend computation using absolute instead of relative anomalies: the results, here not presented, show a discrepancies with respect to panel (a) of Fig. 6 within $\pm 1$ % at all altitudes and latitudes.

10    Gebhardt et al. (2014) applied a MLR analysis to an older version of SCIAMACHY data and found over the 2002–2012 period positive trends in the upper stratosphere at mid-latitudes as well as in the tropics but also a strong negative trend in the tropics between 30 and 38 km up to -10 to -15 % per decade. Other studies have shown similar negative ozone changes in this altitude region. Kyrölä et al. (2013) found an ozone decrease of -2 to -4 % for 1997–2011 using merged data from SAGE II and GOMOS (Global Ozone Monitoring by Occultation of Stars), Eckert et al. (2014) found similar changes in 2002–

15    2012 using MIPAS (Michelson Interferometer for Passive Atmospheric Sounding) observations and Nedoluha et al. (2015) showed a decrease of -0.06 ppmv yr$^{-1}$ using HALOE (Halogen Occultation Experiment) and MLS measurements. Stiller et al. (2017) attributed the changes in the stratospheric age of air (AoA) in the 2002–2012 period to the shift of sub-tropical mixing barriers, which also affects the calculation of long-term changes of stratospheric trace gases. Recently, Galytska et al. (2019) have studied this altitude range in the tropics using SCIAMACHY observations and a run of the Toulouse Off-line Model of

Chemistry And Transport (TOMCAT) chemical transport model (CTM). Model simulations reproduced the observed behavior in the period 2004–2012. They found anti-correlated changes in ozone and $NO_2$ from both SCIAMACHY observations and CTM calculations. They showed that these chemical changes are dynamically controlled by seasonal variations in AoA and thus in the vertical velocity of the BDC. In particular, the CTM showed a slow-down of the vertical transport during autumn months

followed by a speed-up during winter months, causing changes in the residence time of $N_2O$ and as a consequence in $NO_2$ and ozone profiles. When averaged over the whole year, AoA trends of different signs cancel out, resulting in no significant annual mean change, whereas the responses of $N_2O$ and as a consequence $NO_2$ and ozone remain, due to a non-linear relation between chemistry and transport (Galytska et al., 2019). This explains the annual mean trends in the SCIAMACHY ozone profiles observed in the tropical middle stratosphere until 2012. Model studies for the 2003–2018 period are ongoing.

Comparing the results presented in the above mentioned studies with Fig. 6 we notice that the negative trends found in the tropics around 35 km are not detected anymore. To investigate this behavior, ozone time series are displayed in Fig. 7. The debiased time series are plotted along with the full regression fit and the linear trend terms, for the entire time series and the 2003–2011 period. At 34.8 km, panel (a), we notice a decline in ozone until 2010–2012 (with a value close to -10 % per decade), whereas after 2012 the ozone amount in this region returned to values recorded in mid-2000, resulting in nearly no

change in ozone. This fact is enhanced by the anomalous QBO event that occurred in 2015–2016 (Newman et al., 2016), which led to higher ozone in the tropical region during 2016 (Tweedy et al., 2017). In addition, the switch between SCIAMACHY and OMPS-LP time series and the interference between the solar proxy and the trend-terms may enhance this discontinuity in the long-term changes. We have to notice that in this region the SC, as reported in Fig. 2, shows a particularly strong variation between SCIAMACHY and OMPS-LP periods: as a consequence, we found a strong sensitivity of the merging procedure for

anomalies to the period over which MLS SC is computed.

    In the lower tropical stratosphere, panel (b) of Fig. 7, the trend over 2003–2018 is also close to zero. However, looking at the period before and after 2011, we notice that over the SCIAMACHY time a positive trend is present (7 % per decade), which was already reported by Gebhardt et al. (2014). Over the OMPS-LP period the tendency becomes flat or slightly negative, reducing the overall trend.

Focusing on the altitudes where the ozone recovery is identified, panel (a) of Fig. 8 illustrates the latitudinal and longitudinal structure of the long-term ozone changes at 41.3 km, using the anomalies data set within $\pm$ 70° latitude. The longitudinal variability is large, especially in the extra-tropical regions. For example, at northern mid- and high-latitudes ozone changes peak at above 6 % per decade over Canada but are non-significant and around 1–2 % over Siberia. Above Antarctica the trend is also positive, but a dedicated study focusing on ozone distribution during Antarctic spring is needed to assess the on-going

ozone recovery in this region. The longitudinal patterns found in the drift plots and discussed at the end of Sect. 3 do not explain the variability seen in Fig. 8. In order to study the vertical consistency of these longitudinal structures, we show in panel (b) of Fig. 8 the cross section of the trends field at 60° N. We notice that the significant positive values found especially between 180° W and 20° W are vertically homogeneous over three grid levels, from 38 till 44 km. At eastern longitudes the values are consistently non-significant over the whole profile. In the Supplements, Fig. S6 shows in panel (b) the same cross

section at 60° S, where the longitudinal structure is less pronounced, as can be seen also from Fig. 8 panel (a), but it still

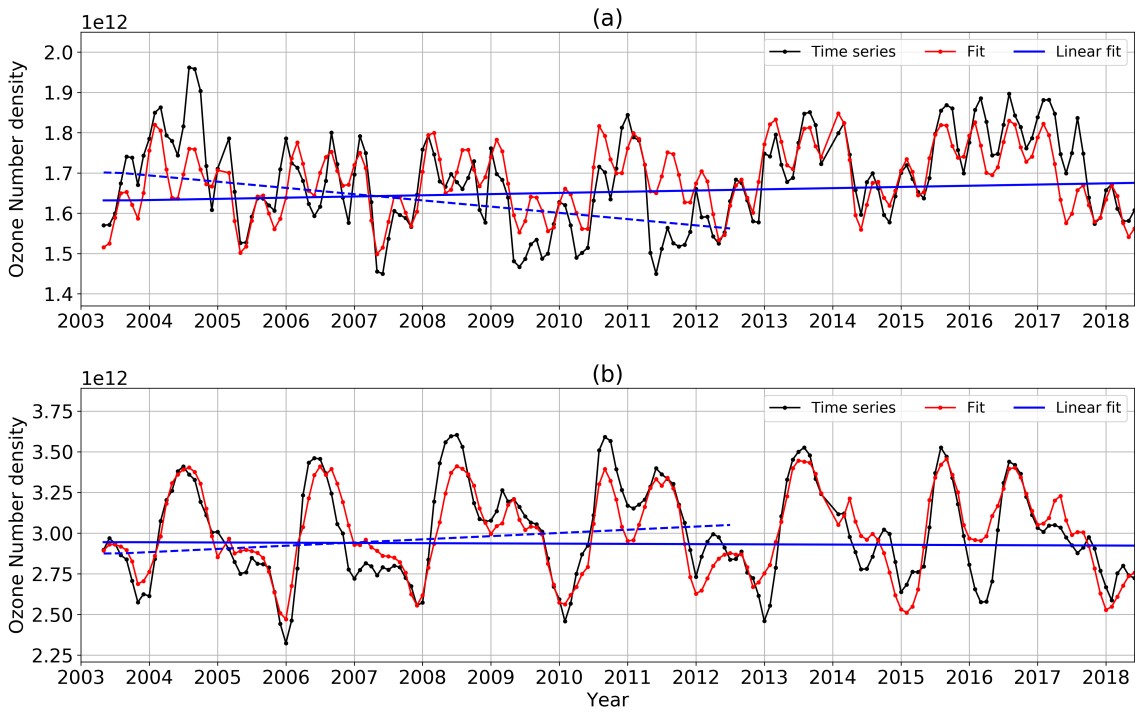

**Figure 7.** Merged number density [molec cm$^{-3}$] time series of SCIAMACHY/OMPS-LP merged ozone ('plain debiased') from 2003 to 2018 (in black) with MLR fit (in red) and linear trend term superimposed in blue (the dashed lines refer to the SCIAMACHY period until early 2012): in panel (a) at 34.8 km and in panel (b) at 21.7 km in the tropics, i.e. [-5°, 5°] latitude bin.

displays a good vertical consistency. Panel (a) of the same picture shows the cross section in the tropics, where values are mostly non-significant and the longitudinal variability is within 1–2 %.

Kozubek et al. (2015) presented the structure of the BDC as a function of longitude and its impact on the ozone distribution, using reanalysis data. At 10 hPa a two-core structure of opposite meridional winds have been clearly identified by the authors
5 at higher northern mid-latitudes, one centered over the Canadian and the other over the Asian sectors. Investigating trends in meridional wind at these heights, they found significant trends in these region, showing a weakening of the two-core structure after the ODSs turn-around point in 1997. These changes in the dynamics of the stratosphere impact the ozone distribution in this region as well. This illustrates the limitations of the zonal mean approach to describe stratospheric dynamics and related ozone trends. Bari et al. (2013), studied the 3D structure of the BDC comparing a general circulation model and MLS
10 observations. The authors found zonal asymmetries in the meridional mass transport, affecting also the ozone and water vapor distribution, particularly in the northern middle winter stratosphere.

Similar maps showing the longitudinally resolved ozone field at 21 km and at 35 km are reported in the Supplements (Fig. S4 and S5, respectively). In the lower stratosphere, we find the already described negative trends in the tropics and

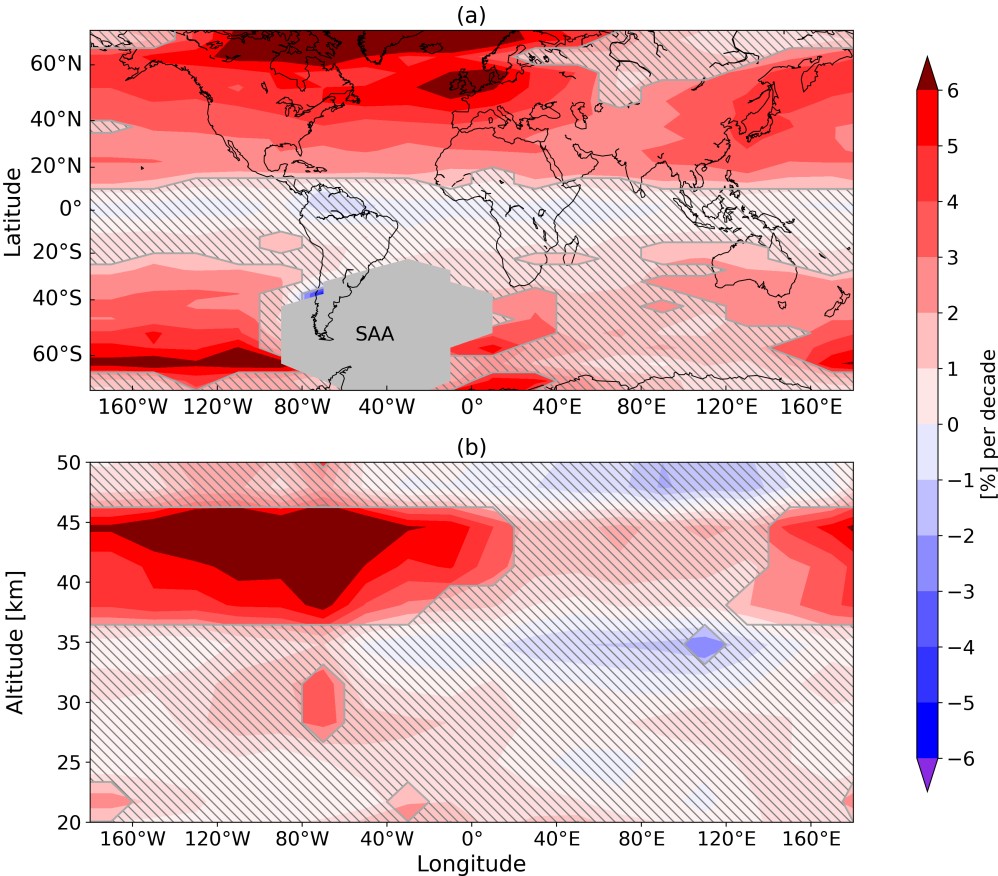

**Figure 8.** Panel (a): longitudinally resolved ozone trends at 41.3 km in % per decade, computed over the 2003–2018 period from the SCIAMACHY/OMPS-LP anomalies merged data. Dashed areas indicate non-significant trends and the gray polygon labeled as SAA indicate the location of the South Atlantic Anomaly. In panel (b), cross section of the longitudinally resolved trends at 60° N.

irregularly positive values at mid- and high-latitudes, in both cases mostly non-significant. This is a possible indication of the speed-up of the BDC, which transports more efficiently ozone towards higher latitudes. At 35 km we recognize similar distributions of the values as at 38–42 km, but with significant positive trends only in the southern hemisphere. At northern mid- and high-latitudes a kind of two-cell structure is found, featuring positive values over Europe and Canada and negative over Russia, even though extensively non-significant.

## 5   Merging with SAGE II data set

The merging of SCIAMACHY and OMPS-LP data sets with SAGE II occultation observations is carried out considering zonal averaged monthly values, gridded every 10° latitude. This approach was followed in order to account for the different geometry and sampling of the three instruments. This enables us to extend the SCIAMACHY/OMPS-LP data record back to 1984. The

sparseness of SAGE II data prevents longitudinally resolved consideration or a finer latitude grid. The merging approach is based on anomalies: SAGE II, SCIAMACHY, MLS and OMPS-LP data sets are deseasonalised using their own SCs and then the offset with respect to SCIAMACHY is removed as done in Sect. 3. The debiasing of SAGE II time series with respect to SCIAMACHY is done using the overlapping period between August 2002 and August 2005. In the merging procedure we
reject altitude–latitude bins in two cases:

  – if less then 10 measurements are available in the bin;

  – if the distribution of SAGE II latitudes inside the bin is not representative for the latitude range, i.e. the mean SAGE II latitude ± its standard deviation do not include the center latitude of the bin.

The same MLR model as discussed in Sect. 3 (without harmonic terms) is applied over four periods and the resulting trends
are shown in Fig. 9, between 20 and 48 km and within $60°$ latitudes. In particular, the trends over the two periods are computed independently and not as a piece-wise linear trend. In the two upper panels of the figure the periods 1985–1997 and 1998–2018 are considered, assuming that in 1997 ODSs concentration peaked in the stratosphere. In agreement with the results presented for example by Sofieva et al. (2017) and Steinbrecht et al. (2017), we find negative trends above 30 km before 1997, reaching up to -6 % in the upper stratosphere at mid-latitudes. After 1998 the trends become positive and are significant at
mid-latitudes above 35 km. We don't see in panel (b) negative trends in the tropics at 35 km. This results from the inclusion of the last 18 months of data. When we consider the time series until 2015 or 2016, a significant negative trend of about -2 % per decade is detected in this region. In the two lower panels of Fig. 9 the focus is brought to the SCIAMACHY and OMPS-LP observation periods to see how short-term ozone changes depend on the periods selected in the MLR. In particular, the January 2004–December 2011 and February 2012–June 2018 periods are considered, as shown in panels (c) and (d), covering
approximately an integer number of QBO cycles. Results in panel (c) can be compared with the trends reported in Gebhardt et al. (2014) and Galytska et al. (2019). Consistent with previous studies, we notice strong negative trends in the tropical middle stratosphere and positive significant trends in the southern lower stratosphere and in the upper stratosphere at northern mid-latitudes. The trends shown in panel (d) over the 2012–2018 period show an opposite picture with respect to panel (c) in the middle and lower stratosphere: positive changes in the tropics around 35 km and negative changes at southern mid-latitudes.
Above 35 km extensively significant positive trends are found at all latitudes. These last two panels show that the long-term changes computed over the last 15 years are the result of complex changes in stratospheric dynamics, which occurred over shorter time scales, and the difficulty to disentangle atmospheric variability from long-term trends. To investigate the possible interference between the solar and the trend terms, we calculated long-term ozone changes for the shorter periods (panel c and d) regressing all non-term terms over the longer period 2003–2018 and then performing a linear trend over the 2004–2011 and
2012–2018 periods. The results for the 2004–2011 ozone changes are shown in the Supplements, Fig. S7: the trend pattern is the same in both cases but differences are visible in terms of absolute values, with smaller trends when the non-trend terms are regressed over the longer period (panel b).

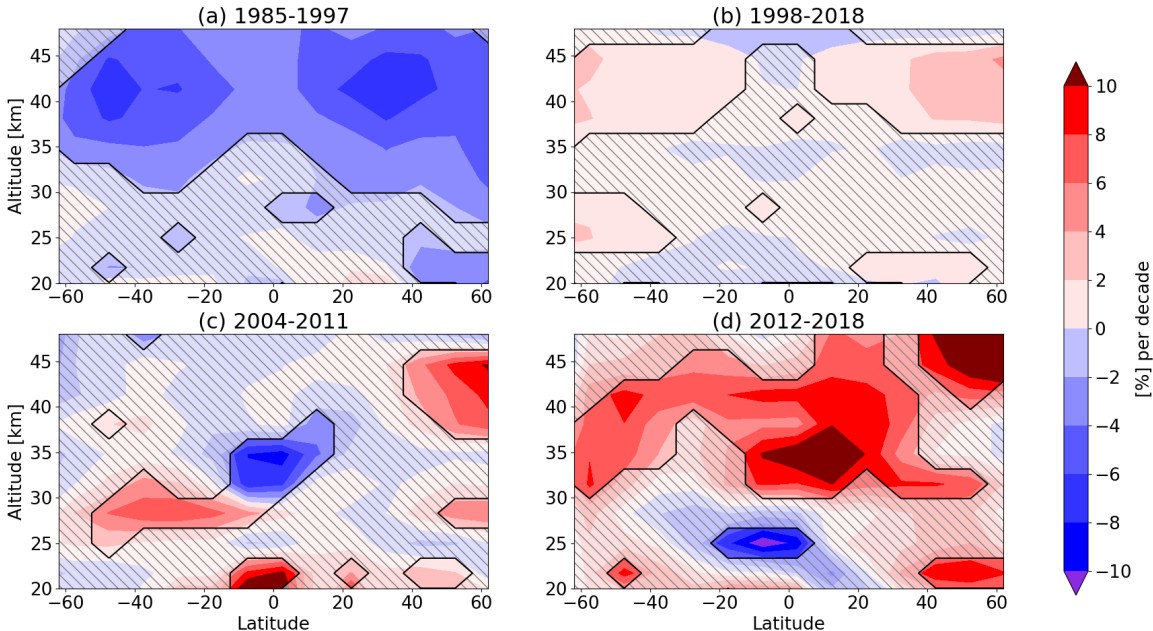

**Figure 9.** Zonal mean trends computed using the merged SAGE II, SCIAMACHY and OMPS-LP data set, in panel (a) over the 1985–1997 period, in panel (b) from 1997 to 2018, in panel (c) over 2002–2012 (SCIAMACHY observation period) and in panel (d) over 2012–2018 (OMPS-LP observation period). Dashed areas show non-significant trends.

## 6 Conclusions

In this paper we described the approach and results of merging SCIAMACHY limb ozone profiles with OMPS-LP measurements. Monthly averaged data have been considered, binned every 5° latitude and 20° longitude, from January 2003 until June 2018. The merging has been achieved using MLS ozone profiles as a transfer function by following two approaches: in the first one ozone number density profiles are directly merged accounting for the bias between the two instruments independently at every latitude, longitude and altitude; in the second the SC of each instrument is first removed and debiased anomalies are then merged. The latter approach is a standard procedure followed in many studies when merging several data sets; in this case, since SCIAMACHY and OMPS-LP observe the atmosphere with a very similar sampling and geometry (in terms of scattering angle), we showed that the 'plain-debiasing' approach is also valid, with the advantage of providing a merged time series expressed in terms of ozone number density and preserving the original SC. Comparing the merged time series with MLS, we found residual seasonal features using the first approach, preventing reliable results at high-latitudes. In the second approach, the merged and MLS anomalies showed discrepancies within ±5 % including the polar regions. A correlation coefficient above 0.8 with respect to the MLS time series and no significant drift between 20 and 50 km and between -70° and 70° latitude pointed out a good consistency of the merged data set.

A MLR model has been applied to the merged data set to study long-term changes in the ozone profile, accounting for several factors affecting stratospheric ozone. Zonal mean trends showed a positive recovery of ozone at mid- and high-latitudes above 35 km, with significant positive changes of about 2–3 % per decade from 2003 until early 2018. Negative but non-significant trends were found in the lower tropical stratosphere. Exploiting the high-spatial resolution of the data set, we also studied longitudinally resolved ozone changes, finding in the middle and upper stratosphere a remarkable trend pattern. This is an indication of a possible change in the BDC as a function of longitudes in the northern hemisphere. A comparison of our results with ozone long-term zonal trends reported in previous studies showed a general consistency with regard to the apparent ozone recovery in the upper stratosphere. However, a change in the sign of the trends in the tropical region over the last 15 years was detected: the strong decrease around 35 km found for example by Gebhardt et al. (2014) over the SCIAMACHY period has vanished when adding nearly five years of data. This is a consequence of several facts, such as the anomalous QBO event in 2015–2016, which led to positive ozone concentration anomalies in the tropics, and a possible change in the stratospheric dynamics with respect to the last decade. Galytska et al. (2019) has recently explained the feature observed over 2004–2011 in terms of a slow down of the BDC during autumn months. At this stage, we hypothesize that the BDC has increased and compensated the previous loss during recent years. Although we have now identified this fluctuation we have not yet unambiguously found its dynamical origin. A model study is planned to improve our understanding of the dynamical impact on the ozone trends in the tropical region over the last 15 years and the variations in ozone concentration over the SCIAMACHY and OMPS-LP periods.

We showed that the differences in terms of trends using the two merging approaches are generally negligible, even though the merging procedure may affect the trends especially in regions where the SC of the instruments showed significant changes over the considered period. As a consequence, we don't see in our case strong advantages using one of the two merging strategies in terms of ozone trend results. Users needs should guide the choice of one of the two merged data sets.

We also studied the impact of MLS conversion from VMR to number density profiles on the computed ozone trends, using pressure profiles from ECMWF ERA-Interim and from MERRA-2 data sets. The results shown in the Appendix, suggested that no significant impact on long-term ozone changes are related to this conversion of the transfer function, with differences within -0.25 and +0.5 % at most altitudes and latitudes. However, the change of the reanalysis database have a non-negligible effect on the MLS profiles themselves, with difference up to 3–5 % above 55 km.

The merging of monthly zonal mean anomalies of SAGE II with SCIAMACHY and OMPS-LP data sets was performed to facilitate the study of zonal trends in particular over the periods 1985–1997 and 1998–2018. We obtained results in agreement with previous studies: decreasing trends up to -6 % per decade at mid-latitudes in the upper stratosphere before the ODSs turnaround point and an upper stratospheric recovery of about 3 % per decade after 1997, as a result of the implementation of measures agreed in the Montreal Protocol and its amendments.

## 7 Data availability

L3 data sets for SCIAMACHY and OMPS-LP as well as the merged time series are available upon registration at the following link: http://www.iup.uni-bremen.de/DataRequest/

## 8 Appendix A

We discuss here the impact of using different reanalysis databases to convert MLS ozone profiles from VMR vs. pressure into number density vs. altitude, as requires for the merging of SCIAMACHY and OMPS-LP time series. In particular, we use two reanalysis databases which provide pressure information needed for the conversion: ECMWF ERA-Interim and MERRA-2. In both cases temperature profiles are taken from MLS observations. In Fig. A1 the effects of this different conversion are displayed. In panel (a) we plot the relative differences as a function of latitude and altitude between the zonally averaged MLS
ozone profiles over 2016, converted using the two databases. In detail, the quantity $MLS_{Rdiff}$ is shown:

$$MLS_{Rdiff}(\phi, z) = \frac{MLS_{ecmwf}(\phi, z) - MLS_{merra}(\phi, z)}{MLS_{ecmwf}(\phi, z) + MLS_{merra}(\phi, z)} * 200 \tag{13}$$

where $MLS_{ecmwf}$ and $MLS_{merra}$ stand for the yearly zonally averaged MLS ozone distributions converted into number density vs. altitude using ERA-Interim and MERRA-2, respectively. Using pressure profiles from ECMWF leads to slightly higher ozone values going towards the upper stratosphere, with a systematic bias up to 3–5 % above 55 km.

We are also interested in the impact that such a conversion has on ozone trends. Indeed, MLS time series converted using the two different reanalysis data sets serves as a transfer function for the merging of the SCIAMACHY and OMPS-LP, so that eventual drifts in one or both reanalysis would lead to some artificial trend. Panel (b) of Fig. A1 shows the differences as a function of altitude and latitude between the 2003–2018 trends computed from the merged 'plain-debiased' SCIAMACHY/OMPS-LP data set, when the MLS time series is converted using the two different reanalysis. As expected the differences in the trends
are small but not negligible in the upper stratosphere: values are within -0.25 and +0.5 % per decade at most of the altitudes and latitudes but approach +1 % per decade above 45 km at some latitudes.

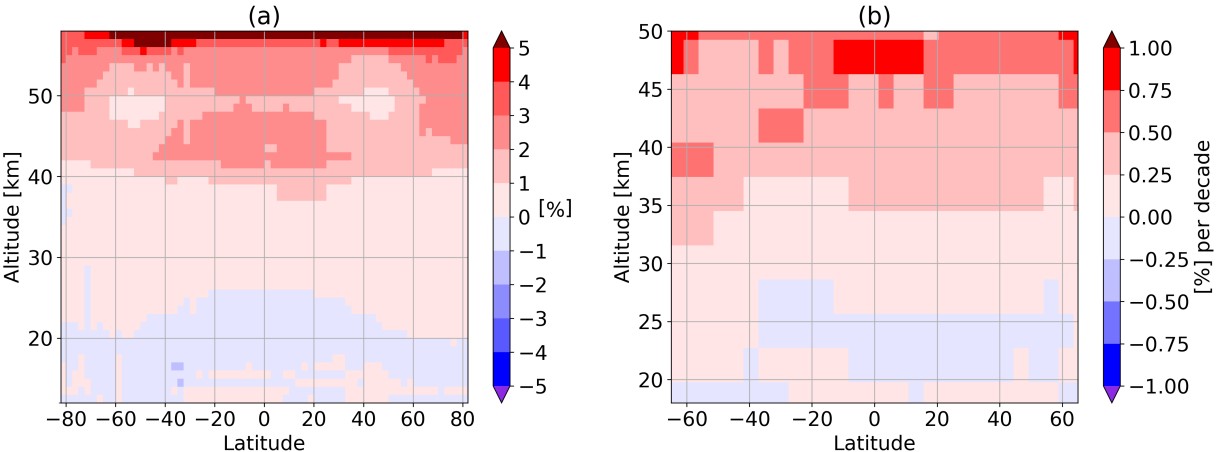

**Figure A1.** Panel (a): relative differences between MLS ozone profiles zonally averaged over 2016 in case the pressure profiles for the conversion from VMR vs. pressure to number density vs altitude are taken from ECMWF or from MERRA-2 reanalysis. Panel (b) effect of using MERRA-2 pressure profiles instead of ECMWF to convert MLS in terms of ozone trends over 2003–2018.

*Author contributions.* CA provided OMPS-LP data set, performed the merging of the time series, developed the procedure to compute trends and wrote the paper. AR supervised and guided the merging and the computation of the trends, provided SCIAMACHY data set and the tools for the MLR model and reviewed the manuscript. EM provided stratospheric aerosol data for OMPS-LP retrieval and revised the manuscript. MW contributed with the fit proxies for the MLR, with the revision of the manuscript and guided SAGE II data handling. JPB initiated and

5   proposed the research and led the project, he contributed to the data analysis, the establishment of the scientific outcomes and the preparation of the manuscript.

*Competing interests.* The authors declare that they have no conflict of interest

*Acknowledgements.* This work was partially funded by ESA within the Ozone-CCI project and was supported by the University and State of Bremen. We would like to acknowledge the NASA team for providing OMPS-LP L1G and Aura MLS data and support. Part of the data

10   processing has been done at the German HLRN (High Performance Computer Center North).

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
