# Peer review of "Merging of ozone profiles from SCIAMACHY, OMPS and SAGE II observations to study stratospheric ozone changes"

_Atmospheric Measurement Techniques, 2018_

## Referee Comment (RC1) · Anonymous Referee #2 · 4 Nov 2018

The manuscript presents two interesting new merged satellite data sets of ozone vertical distribution, based on SAGE II, SCIAMACHY and OMPS observations. Two different methods are used for merging the data sets. The first one uses MLS data as a transfer function to evaluate the bias between SCIAMACHY and OMPS data sets, which overlap for only 2.5 months, while the second merges deseasonalized anomalies. Both data sets have the advantage of being longitudinally resolved, which is generally not the case for similar merged records except for the SWOOSH data set. Yet, the deseasonalized anomalies record when extended with SAGE II observations is zonally averaged. Ozone trends are then computed from the merged data sets using classical multilinear regression over the 2003 – 2018 and 1985 – 2018 periods for the

[Figure]

SCIAMACHY – OMPS and SAGE II – SCIAMACHY – OMPS records respectively. The paper is well written and reference to previous work is adequate. It is suitable for publication in AMT provided that following important comments and recommendations are taken into account.

Major comments

1. The paper is lacking an assessment by the authors themselves of which SCIAMCHY – OMPS merged record they think is best suited for their initial objective of ozone trend evaluation. Comparisons are displayed with MLS data in Figure 3 and 4 of the article, but this record is used as transfer function in both records. What is the advantage for potential users to of using one record over the other one?

2. An assessment of both records could be provided by comparing them to other independent merged records that have been produced recently, e.g. GOZCARDS, SWOOSH, and others.

3. The issue of diurnal variation of ozone deserves some more attention in the article. It is mentioned in page 9 that diurnal ozone variation has to be accounted for above 50 km. However, Sakazaki et al (2013) found significant diurnal variation of ozone well below 50 km and down to 30 km in some latitude ranges.

4. More precision is needed on the use of MLS as a transfer function for both records. What is the processing of MLS data in equations 1 and 2? Are they interpolated to the location of SCIAMACHY and OMPS observations? Similarly, not enough attention is given to differences in vertical resolution between the various data sets. Could it be an issue for the merging? In addition, ERA-Interim is used for the MLS data conversion to number density versus altitude. Did the authors test the sensitivity to other reanalyses such as MERRA2?

Specific comments

P2-l7: The CFCs have been banned by 2010 in Article 5 developing countries.

P2-l23: Sentence starting with N2O is a long-life GHG needs to be rephrased.

P3-l13: What is the reference for the NASA LORE/SOLSE instrument?

P3-l20: Mention instruments on board SCISAT that use the solar occultation technique.

P3-l24: It is an improper description of the Harris et al. (2015) paper. In this paper, merged satellite records are used for trend studies but the merging is not made by the authors. Intercomparison of the merged records is made in Tummon et al. (2015).

P3-l26-27: In general, provide trend results from published studies with error bars.

P3-l30: Ozone-CCI is not the name of a record but the name of a project. More generally in this paragraph it would be better to distinguish articles that describe merged records with those retrieving ozone trends from those records.

P4-l1: Mention the name of the method used in Ball et al. (2018).

P4-l23: The SWOOSH record is resolved in longitude.

P4-l24: Sentence starting with 'In addition': Explain why it is better not to extract the seasonal cycle.

P5- Table1: typo on the unit of the spectral resolution. The latitude coverage could be added in the table as additional information.

P6-l10: A short summary of validation results of OMPS and SCIAMACHY should be added here.

P7-l15: Figure 1 does not include altitude information.

P8-Figure 2: at 28.3 km in the 40°S-20°S latitude range, the SCIAMACHY seasonal cycle looks very different. Can the authors comment on this discrepancy?

P9-l8-9: Equations 1 and 2 should include indices linked to latitude, longitude and altitude.

P13-l6: Sentence starting with 'For the 50-60°N latitude': please clarify. Why is seasonal variation handled differently in this latitude range?

P13-l14: Several studies are mentioned but only one (Park et al., 2017) is cited.

P13-l21: The solar cycle is also used as a proxy in MLR regression of total ozone for trend retrieval in various studies including that from Weber et al. (2017). The solar activity has thus an impact on ozone also in the lower stratosphere. This is worth mentioning.

P17-Fig. 8: Mention for which merged data set are the trends displayed. Trend results should be restricted to the range of validity of the data.

---

## Referee Comment (RC2) · Anonymous Referee #1 · 7 Jan 2019

Review of 'Merging of ozone profiles from SCIAMACHY, OMPS and SAGE II observations to study stratospheric ozone changes', Arosio et al, AMTD, 2018

**1   Short resume**

The construction of long-term ozone profile data records for trend studies has been a lively line of research these past few years. The objective is simple, yet has proven hard to realise: obtain a climate data record which is sufficiently stable (better than a few % per decade) over multiple decades and which ideally covers (most of) the globe at

high spatial resolution. Arosio et al. explore two established, complementary merging methods to combine measurements by two dense limb samplers SCIAMACHY (2003-2012) and OMPS-LP (2012-now) using a third limb sensor as transfer standard, Aura MLS (2005-now). Contrary to most earlier efforts by other groups, the authors attempt to preserve longitudinal information. The resulting data record is then analysed for trends over the 2003-2018 period using a widely used regression model. The authors discuss the spatial structure of the trends and they claim that longitudinal patterns are discerned that are indicative of changes in the Brewer-Dobson circulation. The paper concludes by extending the SCIAMACHY-OMPS data record to earlier decades using similar techniques and SAGE II measurements. Zonally averaged trends from the longer record confirm earlier findings for the 1985-1997 and 1998-2018 periods.

**2 Recommendation**

This paper fits the scope of AMT and would be suitable for ACP as well, since equal shares of the manuscript are devoted to merging methods and trend analysis results. I would recommend publication as long as the authors are willing to improve the discussion of several topics.

**3 Major comments**

3.1 Demonstrate that the longitudinal structures are realistic

My main criticism on this paper is that there is no substantial proof of the robustness of the reported longitudinal structure of the time series and derived trends.

A much more profound discussion is needed about the validity of the longitudinally-

AMTD

resolved results, especially since this is one of the central points of the paper. Constructing a lon-resolved data record is one thing, but the authors need to show that the longitudinal information in the data record is reliable and stable. This should have been the cornerstone of this paper, but it is entirely missing from the paper.

As an illustration (p.12, l.2): "A plot of the longitude-resolved drift values is shown in the Supplements, Fig. S1". But no discussion of key results follows: is there longitude structure in the drift field, or not? Another check would be to compare lon-resolved maps of trend results at neighbouring vertical levels to demonstrate their stability in the vertical domain. Once this validation step is over with, you could gain additional confidence by discussing how the derived trend fields compare to what is expected.

**3.2   Elaborate discussion of merging technique**

The authors present two merging methods and the resulting difference time series with respect to Aura MLS (Figs. 3-4). Unfortunately, they miss the opportunity to discuss merits and weaknesses of each of the methods and in what way one or the other method can correct for specific issues. I feel such a discussion in Sect. 3 would improve the paper a lot. In the end, readers of this paper will be interested in what you recommend as merging approach: plain-debiasing or anomalies? The answer to this naive question may depend on the use case, of course, but this should be part of the discussion.

For instance, Fig. 4 shows a discontinuity of in the anomaly-merged time series between 10°N-10°S at 31.5 and 34.8 km. What is the cause of the feature and why is it not present in the plain-debiased time series (Fig. 3)? The trends in the tropics (Fig. 6) are, surprisingly, not very different using both data records. How can this be? On the other hand, p.14, l.14 claims "The general picture in the two panels is very similar, even though trend values in panel (b) are slightly larger". Can this observation be linked to the merging strategy?

**3.3 Absolute vs relative offset corrections**

The adopted plain-debiasing method (Eqs. 1-2) removes additive biases but not the multiplicative biases. And vice versa, the adopted anomalies method (Eqs. 3-4) removes multiplicative biases but not the additive biases. Can the authors clarify the statement on p.8, l.8-9: "Through the merging process biases will be subtracted, whereas the discrepancies in the shape of the SC are accounted for when calculating anomalies (subtraction of the SC)."?

**3.4 Substantiate claim about stability MLS seasonal cycle**

p.8, l.3-4: "In addition, we notice that MLS SC may vary within the instrument life time, as shown at 34.8 km in the tropics with change of up to 5–7 % between the two periods." This is quite a bold statement which may worry the users of Aura MLS data. But this claim is not really substantiated by the authors. Should a reader be really worried about the stability of MLS data, while you mentioned earlier on that it is stable? When looking at Fig. 2 only one panel indicates that MLS 2005-2012 deviates clearly from MLS 2012-2016. I would like to see more proof/discussion if you want to keep the statement that MLS SC varies over its life time.

**3.5 Collapse of longitude dimension**

p.7, l.15: "In this paper we only describe the analysis of the longitudinally resolved ozone profile product". If you do not consider the zonally averaged data in this Section, why mention the binning at all? This is confusing as most of the plots in Sect. 3 are latitudinal cross sections. More importantly, the authors do not clarify in what order and how the different dimensions are collapsed from the underlying alt-lat-lon-time resolved data, perhaps because it has no importance but -in that case- it should be

stated somewhere. For e.g. Figs. 3 and 4, did you collapse longitude dimension before computing the difference to MLS, or, first compute difference to MLS then average over longitude?

**3.6 Diurnal variation**

p.9, l.24-25 reads "Furthermore, at these altitudes diurnal variation of ozone have to be accounted for (Sakazaki et al., 2013), which was not done in this study". This message is repeated on p.15, l.5-6. The correction scheme Eq. 1-2 removes (additive) biases between data records, irrespective of the nature of the bias. Biases due to diurnal variation are part of the total bias. Hence, I infer that diurnal variations are accounted for contrary to what the authors claim. Can the authors respond to this reasoning, and incorporate their answer in the manuscript?

**3.7 Impact of using ERA-Interim data to convert Aura MLS data?**

The authors mention that the Aura MLS data record is stable (p.6, l.18-19). However, it is not clear from the paper whether this holds for converted Aura MLS data as well. Please elaborate on how the ERA-Interim data may impact the converted Aura MLS data. Can it induce the change in seasonal cycle reported on p.7, l.6-7? Can it lead to the drift above 50 km reported in p.9, l.22-23?

**3.8 Correlation between solar and trend term**

The MLR regression model contains a term for the 11 year solar cycle and a linear trend term (p.12, Eq.6). The analysis period (2003-2018) contains one and a half solar cycle, which triggers the question as to how independent the two said low-frequency terms are. Can the authors elaborate on this? Could the change in derived trend for

different starting times (p.15, l.7-10) be related to interference between the solar and trend term, this is exactly the region where solar influence should be large.

This concern may even be more important for the results shown in Figs. 7 and 9 where even shorter periods are regressed. Perhaps in these cases the non-trend terms were regressed over the entire time period?

**4 Minor comments**

p.1, l.11: Be specific about what you mean with "remarkable variability".

p.2: Very nice and concise overview of ozone-related processes.

p.3, l.19: Identify "MLS" as "Aura MLS" here and throughout the rest of the paper. You don't want to confuse with the first MLS instrument which was flown in the 1990s-2000s on the UARS satellite.

p.3, l.19-21: You should introduce SAGE II over here, instead of two instruments (ACE-FTS and SAGE III/ISS) that are not mentioned in the rest of the manuscript.

p.3, l.25: Please rephrase. Harris et al (2015) did not merge these data sets, but use them to derive trends.

p.3, l.34: Remove "applying a multilinear regression analysis". This information is evident and not different from the other analyses you refer to.

p.4, l.1: Vague statement "Ball et al. (2018) applied a method independent from the ozone turnaround point". The subsequent clause "showed for the first time some evidence of a negative trend in lower stratospheric ozone" seems to imply that the different regression method is leading to this discovery. I am not sure that is what Ball et al. claimed.

p.4, l.4-5: Hanging statement: "This analysis has recently been challenged by Chipperfield et al. (2018)". In what way?

p.4, l.6: Clarify what a "pointing drift" is. A general reader will not have a clue what pointing means in this context. Consider vertical pointing, altitude registration, ...

p.4, l.15: I am not sure LOTUS is "homogenizing" the merging procedures, please double check this with one of the LOTUS participants.

p.5, l.12: "separated by 4.25deg at the tangent point." I assume you mean the viewing angles differ by 4.25deg.

p.5, Tab. 1: Extend this table to SAGE II and Aura MLS.

p.5, Tab. 1: Add level 2 versions in this table as well. This will make life easier for readers in 5 years from now.

p.5, Tab. 1: I advise to show the analysis time period for both instruments. Right now, different information is conveyed in "data time series": SCIA (full mission period) and OMPS (analysis time period). Please use one or the other, but do not mix up.

p.5, Tab. 1: Align values of spectral coverage and spectral resolution with what is in the main text.

p.5, l.20: Add the version of SCIATRAN.

p.6, l.1: Add the version of the SCIAMACHY L1 data, as was done for OMPS-LP.

p.6, l.9: Clarify "pointing knowledge issues", see also my earlier comment. E.g. "[...] when the issues related to the vertical pointing of the instrument, currently under [...]"

p.6, l.9: Refer to Moy, AMT 2017.

p.6, l.13: Replace "scientific measurements" by a better description or simply drop "scientific".

p.6, l.18: You mention Hubert et al. (2016) for Aura MLS drift. But any trend paper should refer to published drift results for all instruments involved. I.e. add those for SCIAMACHY (Rahpoe-2015, Hubert-2016, LOTUS-2018, ...?) and OMPS-LP (Kramarova-2018) as well, unless the L1-L2 versions have changed sufficiently to question the validity of those values.

p.6, l.18: Find a better phrasing for "For technical reasons" as it suggests an instrument malfunction. The observations by SAGE II are sparse due to the chosen measurement geometry and is unrelated to the instrument itself.

p.6, l.31: "Aura MLS", see earlier comment.

p.6, l.33: "taking only the latitude covered daily by OMPS-LP". You lost me here, what latitudes are not covered by OMPS? Please clarify what you mean in the main text. And why does this resolve an inconsistency? SCIAMACHY measurements are also made during daytime.

p.7, l.2: "Aura MLS", see earlier comment. Please incorporate this comment in the rest of the manuscript.

p.7, l.3: Add a motivation for not using the 2002 SCIAMACHY data.

p.7, l.11-12: Figure 1 does not confirm the statement "In both cases we find about 100 profiles on average in each bin." for SCIAMACHY. Each monthly 5deg zonal bin has 1000 profiles, which translates to 56 (=1000/18) profiles per bin, not 100. Did I misunderstand? If not, please change the misleading statement.

p.7, l.14: Clarify what interpolation method was used.

p.7, Fig.1: Add 5° at the end of the caption: "[...] in each 5° zonal monthly bin [...]".

p.8, Fig.2: Is the SCIAMACHY time period identical to that of Aura MLS (2005-2012)? Please add the time period for all four lines in the legend, not just for Aura MLS.

p.8, l.5-6: The phrasing is not clear whether the time period was only adapted for MLS. In other words, did you compare 2005-2012 for both SCIAMACHY and MLS, and

2012-2016 for both OMPS-LP and MLS? See previous comment.

p.9, l.14: Add a short phrase that the unit of the plain-debiasing data set is ozone number density.

p.9, l.16: "[...] differences between the merged data set [...]". What merged data set? The zonal one? The longitudinally resolved one?

p.9, l.16-18: What is the sign of the relative difference? (SCIAOMPS - MLS) / MLS or the other way around?

p.9, l.28-29: Are the larger relative difference values at 15 km truly due to lower data quality or due to the smaller number densities in the UTLS region?

p.9, l.30-31: Replace by "[...] deseasonalized relative anomalies from [...]" to clarify that you are not working with absolute anomalies.

p.9, l.30-31: What is the motivation behind debiasing the deseasonalized relative anomalies? By computing the anomaly any multiplicative biases are removed by definition.

p.9, l.31: Replace by "[...] month of the year, m, the (relative) anomalies, [...]".

p.10, Fig. 3 and p.11, Fig. 4: Add in the colour scale the exact sign of the difference: (merged - MLS) / MLS or (MLS - merged) / merged ?

p.10, Fig. 3 and p.11, Fig. 4: Add in the caption that MLS data has been offset to SCIA prior to the comparison.

p.10, l.4: Eq. 5 is not really used in the rest of the paper (p.12, l.7-8). I would therefore suggest to drop it, also because (a) you do not explain how the uncertainty for the plain-debiasing data is computed and (b) there is no term for the uncertainty in the seasonal cycle.

p.10, l.9: Replace by "Figure 4 shows the absolute differences [...]" to clarify that these

are absolute differences of relative anomalies.

p.11, l.1: "whereas below 20 km the pattern becomes rather chaotic".

p.11, Fig.4: Larger differences are found around 35 km in 10S-10N during the OMPS-LP period. What is the cause of this? Does the different MLS SC in the two periods play a role?

p.11, l.4-5: "The drift is computed as the linear change of the differences between the merged time series and MLS data [...]". Are these relative differences for plain-biased merged data and absolute differences of anomaly-merged data? Clarify this in the text. And add the unit of the drift: % per decade/year/... .

p.12, l.1-2: "Very similar results for the drift are obtained using anomalies time series". The timeseries in Figs. 3-4 look fairly different in places, and I am surprised the drift results are very similar for the anomaly time series. This plot has to be in the main paper, also since it may be the basis of an interesting discussion on what merging technique led to most stable results for this particular case. (See also one of my major comments.)

p.12, Fig. 5: Add in the colour scale the full unit (% per decade/year/...) and the exact sign of the difference: (merged - MLS) / MLS or (MLS - merged) / merged ?

p.12, l.6: What are the units of O3 in Eq. 6? The plain-debiased time series are in molec cm$^{-3}$, the anomaly-merged time series are in %?

p.12, l.14-15: The phrase "The t-th row of the X matrix contains the values of the fit terms for the selected t." does not add information. It could easily be dropped.

p.13, l.1: The equivalence of the $2\sigma$ rule to 95% confidence level is introductory statistics, hence the reference to (Tiao et al., 1990) is not needed.

p.13, l.3: How are the plain-debiased time series in molec cm$^{-3}$ regressed to obtain % per decade?

p.13, l.6-9: Do I understand you correctly that the EHF term is used instead of the harmonic terms, below 25 km and only for the 50-60°N band? Why not for 50-60°S as well, or at other latitudes? Please add that this modified regression model is not applied to the analysis of anomaly-merged time series.

p.13, l.23: Please cite more recent work, at least Maycock et al. (2016), perhaps others as well (Ball et al., 2016; Damadeo et al., 2018; ...).

p.14, l.1: Add the source of the El Nino 3.4 index data, as you did for the other proxy data sets.

p.14, l.6: Add that $N_34$ represents the El Nino 3.4 index anomaly data.

p.14, l.11-12 and Fig. 6: Are the trends in Fig. 6 regressed directly from the zonally averaged merged time series, or are these trend results regressed from lat-lon resolved merged time series then averaged over the latitude bands? In the first case, this contradicts an earlier statement that only analysis of lon-resolved data would be described (p.7, l.15). In the latter case, how do you propagate the uncertainties?

p.15, l.2: You may consider adding the LOTUS report and replace the WMO reference in this phrase (WMO, 2018).

p.15, l.9: Perhaps the cause is not instrumental, but related to the interference of the solar and trend term? See my major comment above.

p.15, l.35: Are all terms (seasonal, QBO, solar, ENSO, ...) constrained by the 2003-2011 period or just the linear trend term? This shorter period potentially makes the interference between solar and trend terms even larger. Have you looked into this? The trend results may be more stable/robust when you constrain all non-trend terms (especially solar) to the larger 2003-2018 period.

p.16, Fig. 7: Add complete unit to y-axis label: molec cm$^{-3}$.

p.16, Fig. 7: Adding the anomaly-merged time series and fits would make a fine illustration of how the merging strategy can overcome some of the issues in the data sets as mentioned e.g. in p.16, l.4-7.

p.16, l.1: There is the switch to OMPS-LP in 2012. Can this be a viable alternative explanation to the "discontinuity"? The fits themselves will, in addition, likely be impacted by the solar-trend interference as well.

p.16, l.11-12: Please substantiate why the longitudinal trend results are reliable? A figure like Fig. 8 for neighbouring levels z=41 and 44 km will help to demonstrate the stability of the results in the vertical domain, especially in the US where trends are mostly significant (also in other studies).

p.17, Fig. 8: Add to the caption what merged data set was used: plain-debiased or anomaly-based?

p.17, Fig. 8: Remove the results in the latitude-range that you mentioned earlier on was not reliable (60° for plain-debiased and 70° for anomaly-based).

p.17, l.10-11: Move this discussion to previous paragraph and elaborate on how stable results are in vertical domain.

p.17, l.15-16: Motivate why you use the anomaly approach.

p.18, l.6-7: Add brief explanation why the harmonic terms are not included (deseasonalized anomaly time series).

p.18, l.6-7: Slightly confusing, since the trend model is very different from that in previous section. Please clarify whether it is an independent trend (ILT) or a piece-wise trend (PWLT).

p.18, l.13: Add correct unit : "[...] about -2% per decade is detected [...]".

p.18, l.13: As asked before (p.15, l.35), what time period was used to constrain the non-trend terms? And how robust are -especially- the 2012-2018 trend results given the low frequency of the 11 year solar cycle proxy?

p.18, l.19: You mention 2010-2018 here, while the figure caption says 2012-2018. Which one is correct?

p.18, l.32: What do you mean with "up to" the polar regions?

p.19, Fig. 9: You mention 2012-2018 in the caption, while the main body text says 2010-2018. Which one is correct?

p.19, Fig. 9: Add the time period to each panel, in addition to (a), (b), ...

p.19, l.6-7: Strong claim that needs demonstration: how reliable is the observed lon-resolved structure?

Supplement, Fig. S1: Add in the caption which merged time series are shown: plain-debiased or anomaly?

Supplement, Figs. S1 and S2: Add sign and correct unit (% per decade?) to colour scale or in the caption.

Supplement, Figs. S1 and S2: Each subpanel represents one longitude-bin, all together they convey information about longitude structure of drift of SCIAOMPS wrt MLS. However, the longitude structure would be much more obvious if you would have shown one latitude-bin per subpanel, and then plot drift vs altitude vs longitude. Can you add this to the supplement?

Supplement, Figs. S3 and S4: Add to the caption what merged data set was used: plain-debiased or anomaly-based?

Supplement, Figs. S3 and S4: Remove lat-range with data that you claimed earlier in the paper are unreliable (poleward of 60° or 70° latitude).

**5 Technical corrections**

p.1, l.4-5: Replace by "[...] is performed by the processor of the University [...]"

p.1, l.10: Replace "high" horizontal sampling by "dense" horizontal sampling.

p.1, l.24: Remove either "important" or "key". Important implies key and vice versa.

p.2, l.7: Replace by "during the 1990s".

p.2, l.10: As non-native speaker I expected "GHGs such as", but perhaps "such" is not needed.

p.3, l.27: Add a "," in "[...] before 1998, and a positive trend of [...]".

p.3, l.29: Replace by "[...] with uncertainty estimates. [...]".

p.3, l.30: Add "-", replace by "[...] satellite and ground-based data sets [...]".

p.3, l.35: Replace by "[...] significant trends [...]".

p.3, l.35: Replace by "[...] in the upper stratosphere at mid-latitudes [...]".

p.5, l.4: Replace "[...] in-flight direction [...]" by "in flight direction" or "in the direction of flight".

p.5, l.13: Remove the first "charged" in "charged charged-coupled device".

p.5, l.22: Replace "application of SCIAMACHY retrieval scheme" by "application of SCIAMACHY's retrieval scheme".

p.6, l.3: Replace "we take into account in addition" by "we also take into account".

p.7, l.23: Replace by "[...] the SC of all single instrument data sets [...]".

p.8, l.3: Replace by "[...] the three ozone profile data records in number density [...]".

p.13, l.16: Replace by "[...] and the in-phase at mid-latitudes [...]".

p.13, l.32: Drop "a" in "[...] leading to longitudinally dependent modifications of ozone [...]".

p.14, l.17: Replace by "Bourassa et al. (2018)".

p.15, l.11: Replace "detected" by "found" or "observed". In my view, "detected" implies that the result is significant which is not the case.

p.15, l.33-34: Remove newline after "[...] panel (a).".

p.17, l.13: Add "s" to "[...] SAGE II occultation observations [...]".

p.18, l.4-5: Replace by "[...] the mean SAGE II latitude plus or minus its standard deviation [...]".

p.18, l.7: Replace by "[...] $60°$ latitude [...]".

p.18, l.29: Replace by "[...] is first removed [...]".

p.18, l.30: Remove "one" from "[...] MLS one [...]".

p.18, l.32: Replace by "[...] with respect to the MLS time series [...]".

p.19, caption: Replace by "[...] and in panel (d) over 2012–2018 [...]".

p.19, l.4: Replace by "[...] from 2003 until early 2018 [...]".

p.19, l.5: Replace "detected" by "found" or "observed". In my view, "detected" implies that the result is significant which is not the case.

p.19, l.10: Replace by "[...] has vanished when adding [...]".

---

## Author Comment (AC1) · 11 Feb 2019

**Replies to Referee #2 on the manuscript 'Retrieval of ozone profiles from OMPS limb scattering observations' by C. Arosio et al.**

We thank the reviewer for the time she/he spent reading the manuscript and constructively commenting on the paper. In the text below, we address the comments from the Referee #2. Referee's comments are shown in italicized font and authors' responses are highlighted in blue.

**Major comments** *The manuscript presents two interesting new merged satellite data sets of ozone vertical distribution, based on SAGE II, SCIAMACHY and OMPS observations. Two different methods are used for merging the data sets. The first one uses MLS data as a transfer function to evaluate the bias between SCIAMACHY and OMPS data sets, which overlap for only 2.5 months, while the second merges deseasonalized anomalies. Both data sets have the advantage of being longitudinally resolved, which is generally not the case for similar merged records except for the SWOOSH data set. Yet, the deseasonalized anomalies record when extended with SAGE II observations is zonally averaged. Ozone trends are then computed from the merged data sets using classical multilinear regression over the 2003 – 2018 and 1985 – 2018 periods for the SCIAMACHY – OMPS and SAGE II – SCIAMACHY – OMPS records respectively. The paper is well written and reference to previous work is adequate. It is suitable for publication in AMT provided that following important comments and recommendations are taken into account.*

**General Comment**
*1. The paper is lacking an assessment by the authors themselves of which SCIAMACHY – OMPS merged record they think is best suited for their initial objective of ozone trend evaluation. Comparisons are displayed with MLS data in Figure 3 and 4 of the article, but this record is used as transfer function in both records. What is the advantage for potential users to of using one record over the other one?*

There are two points raised by the reviewer.
The first one, regarding which SCIAMACHY/ OMPS-LP merged record is best suited for trend studies, was also addressed by Reviewer #1 and doesn't have a simple answer: we showed that both methods give similar results in terms of ozone trends. The advantage of the plain-debiasing approach is that it maintains the original data sets as they are, so that the resulting merged time series is expressed in terms of ozone number density and preserves the original seasonal cycle. This can be useful for example for data assimilation and model studies, for which a time series in terms of number density or VMR is more valuable than ozone anomalies. The second method involves the subtraction of the seasonal cycle and it is more suitable at altitudes and latitudes (like polar region) where the seasonality of the instruments differ more strongly. The subtraction of the seasonal cycle is a common procedure before the merging, especially when considering several satellite data sets with different observation geometry and latitude coverage. In our case SCIAMACHY and OMPS-LP observe the atmospheric scenes at a very similar scattering angle, the latitude coverage is comparable (as reported in Table 1) and the overpass time differs by 3.5 h. As a consequence we believe that the plain-debiasing method is in this case also suitable and reliable. A sentence was added in the conclusions about this point: '[The anomaly] approach is a standard procedure followed in many studies when merging several data sets; in this case, since SCIAMACHY and OMPS-LP observe the atmosphere with a very similar sampling and geometry (in terms of scattering angle), we showed that the plain-debiasing approach is also valid, with the advantage of providing a merged time series expressed in terms of ozone number density and preserving the original seasonal cycle.'
The second point relates to the use of MLS as transfer function and for comparison/validation: since MLS data set was not included in the merged record and no drift correction has been

applied, we think that it can be considered an 'independent' data set for validation. The MLS data in the merging procedure is only used to remove the systematic mean bias between the instruments. After the debiasing procedure, the mean levels of the time series are the same for the 3 instruments (at each altitude, latitude and longitude) but the ozone variability and the seasonal structure remain independent. That's why the computation of correlation and relative differences between MLS and the merged time series is justified.

*2. An assessment of both records could be provided by comparing them to other independent merged records that have been produced recently, e.g. GOZCARDS, SWOOSH, and others.*

Several recent merged data sets such as GOZCARDS and SWOOSH for the period after 2005 are determined by MLS observations. As a consequence, we consider the usage of these data sets for further validation not relevant, since the comparison would lead to very similar results as the ones we got using MLS only. Previous works like Harris et al., 2015 and Steinbrecht et al., 2017 already compared trends from several merged satellite data sets and illustrated the differences.

*3. The issue of diurnal variation of ozone deserves some more attention in the article. It is mentioned in page 9 that diurnal ozone variation has to be accounted for above 50 km. However, Sakazaki et al (2013) found significant diurnal variation of ozone well below 50 km and down to 30 km in some latitude ranges.*

The reviewer is right, Sakazaki et al., (2012) showed that also at 30–40 km daily variations of ozone play a non-marginal role. However, we did not take them into account for the following reasons. First of all, the equatorial overpass times of SCIAMACHY and OMPS are both around the noon, namely at 10:00 and 13:30 respectively. This was not clearly stated in the manuscript and this information has been added in Table 1 and remarked in a sentence in Sect. 3: 'This [considering diurnal variations] was not done in our study, because the equatorial crossing time of the two instruments is around noon and differs by only 3.5 h: this would lead to a systematic discrepancy in ozone that we estimate to be about 1-2 % at 30–40 km. Furthermore, the expected systematic bias between the two instruments is largely removed by the debiasing procedure, even though not completely, because variations with time of this systematic discrepancy may not be accounted for by a 'plain-debiasing' procedure.' Indeed, according to the mentioned paper, at altitudes between 30–40 km the ozone concentration has a minimum in the morning after dawn and a maximum in the afternoon, with an amplitude in terms of VMR of 0.15 ppmv, which corresponds to 2–3 % of the ozone at these altitudes. So, considering the satellite equatorial crossing times we expect a difference in the order of 1–2 %. Anyway, the important point to be stressed is that the effect of such a diurnal variation leads to a systematic effect on the whole time series, since the overpass time remains constant; such an offset is then largely removed by debiasing the data sets. We therefore believe that the debiasing procedure to a large extent eliminates this systematic difference between the two instruments, as long as it remains constant with time.

*4. More precision is needed on the use of MLS as a transfer function for both records. What is the processing of MLS data in equations 1 and 2? Are they interpolated to the location of SCIAMACHY and OMPS observations? Similarly, not enough attention is given to differences in vertical resolution between the various data sets. Could it be an issue for the merging? In addition, ERA-Interim is used for the MLS data conversion to number density versus altitude. Did the authors test the sensitivity to other reanalyses such as MERRA2?*

In Eq. 1 and 2 MLS time series as well as SCIAMACHY and OMPS-LP ones are already considered as monthly mean profiles, binned into the regularly spaced grid in terms of latitude and longitudes. Before computing monthly averages and binning the data, each single MLS profile has been converted into number density vs. altitude and interpolated onto the common vertical grid (equally spaced every 3.3 km). A sentence was added in the paper to clarify this point: 'In these and following equations, ozone profiles from each instrument are considered as binned monthly averages, interpolated to a common altitude grid.'

The issue related to the vertical resolution is interesting but difficult to remove. It has to be noted that the vertical resolution of the three sensors is similar: about 3 km for all instruments. The vertical sampling is however higher for OMPS-LP and MLS, whose ozone profiles are provided every 1 km, and is equal to 3.3 km for SCIAMACHY. This information has been added to the paper. We interpolated all single profiles from each instrument in the same vertical grid using a linear interpolation scheme. This procedure may indeed introduce artificial discrepancies between OMPS-LP and SCIAMACHY, especially at altitudes where the seasonal cycle significantly changes. To minimize this problem, we chose to perform the interpolation of the 1 km-spaced OMPS-LP and MLS profiles onto the SCIAMACHY lower vertically-resolved grid and not vice versa.

The reviewer raises another interesting point here. It was indeed not properly explained in Sect. 3 that for the conversion from MLS VMR vs. pressure profiles to number density vs. altitude, only pressure profiles from ECMWF are considered, while the temperature profiles are taken from MLS retrievals. We updated the description: 'Volume mixing ratio ozone profiles from MLS on a pressure grid are converted to geometric altitude vs. number density using collocated pressure information from the ECMWF ERA-Interim database and temperature profiles retrieved by MLS.' No issues are known by the authors about ERA-Interim pressure, so that we think that the usage of a different reanalysis product would lead to non-relevant changes. However, a comparison in terms of relative differences between MLS ozone profiles converted using ECMWF and MERRA-2 has been performed and the results have been included in the paper, in Appendix A, Fig. A1. Computing the relative difference between number density MLS zonally averaged ozone distributions over 2016 as a function of altitude and latitude, computed using the two reanalysis, we see (panel (a) of the picture) that the discrepancy increases with altitude, up to 3-5 % above 55 km, while in the lower stratosphere differences are within 1 %. In addition, the sensitivity of the ozone trends to the MLS conversion have been studied as well: the right panel of Fig. A1 in the paper reports the differences in terms of % per decade between the trends computed from the merged data set (plain-debiased approach) when using ECMWF or MERRA-2 for conversion. The differences are small even though not always negligible in the upper stratosphere: values are within -0.25 and +0.5 % at most altitudes and latitudes, approaching +1 % above 45 km at some latitudes.

**Specific comments**
*P2-l7: The CFCs have been banned by 2010 in Article 5 developing countries.*

The sentence have been reformulated as: 'the London amendment in 1990 called for a complete phase out of CFCs production by the year 2000, ...'

*P2-l23: Sentence starting with N2O is a long-life GHG needs to be rephrased.*

The sentence has been split in two parts for better readability.

*P3-l13: What is the reference for the NASA LORE/SOLSE instrument?*

Thanks, a reference was included also for LORE/SOLSE: McPeters et al. 2000.

*P3-l20: Mention instruments on board SCISAT that use the solar occultation technique.*

Thanks, instead of mentioning the SCISAT satellite we directly referred to the ACE-FTS and MAESTRO instruments, as we mention SAGE III instrument in the same sentence.

*P3-l24: It is an improper description of the Harris et al. (2015) paper. In this paper, merged satellite records are used for trend studies but the merging is not made by the authors. Inter-comparison of the merged records is made in Tummon et al. (2015).*

We thank the reviewer for this note. The description of Harris et al. (2015) was changed from 'the authors considered several satellite data sets, merged them over the period 1979–2012 and examined separately...' to 'the authors considered several existing merged satellite data sets and examined separately...'

*P3-l26-27: In general, provide trend results from published studies with error bars.*

A better characterization of the values has been provided for each cited work, except for Harris et. al (2015), for which several uncertainties were discussed and it was difficult to summarize the results in few sentences.

*P3-l30: Ozone-CCI is not the name of a record but the name of a project. More generally in this paragraph it would be better to distinguish articles that describe merged records with those retrieving ozone trends from those records.*

Thanks, we changed the terminology regarding Ozone-CCI: from 'merged measurements from SAGE II with Ozone-cci and OMPS satellite data sets' to 'merged measurements from SAGE II with several other data sets homogenized within the Ozone-CCI project including OMPS-LP'. We also improved the distinction between papers merging data sets from articles retrieving trends from the merged data sets.

*P4-l1: Mention the name of the method used in Ball et al. (2018).*

Yes, the name dynamic linear method has been included.

*P4-l23: The SWOOSH record is resolved in longitude.*

Thanks for the note, "except for SWOOSH" was added.

*P4-l24: Sentence starting with 'In addition': Explain why it is better not to extract the seasonal cycle.*

Generally speaking it is better to extract the seasonal cycle to remove discrepancies between data sets related to the sampling, geometry, seasonality and overpassing time; in this case we performed also this approach since we consider only 2 instruments with similar characteristics in terms of geometry and sampling. No further explanation has been added at this point but we expanded the discussion of the two approaches in Sect. 3.

*P5- Table1: typo on the unit of the spectral resolution. The latitude coverage could be added in the table as additional information.*

The table was corrected and the information about the latitude coverage for both instruments included.

*P6-l10: A short summary of validation results of OMPS and SCIAMACHY should be added here.*

We agree with the reviewer. We included at this point of the paper a couple of sentences about the results of SCIAMACHY validation against ozonesondes and IUP-OMPS validation against MLS and sondes.

*P7-l15: Figure 1 does not include altitude information.*

Yes, it doesn't include altitude information because it just shows the number of available (retrieved) profiles from the two instruments as a function of time, without considering it as a function of altitude.

*P8-Figure 2: at 28.3 km in the 40°SS-20°SS latitude range, the SCIAMACHY seasonal cycle looks very different. Can the authors comment on this discrepancy?*

We studied more carefully SCIAMACHY seasonal cycle in this region and the main discrepancies are found at [40°S, 30°S] latitude, where its seasonal cycle is pretty flat after the maximum in February-March, in comparison with MLS seasonal cycle in the same period. We don't know the reason for this difference.

*P9-l8-9: Equations 1 and 2 should include indices linked to latitude, longitude and altitude.*

We included the indexes '(lat, lon, z)' also in equation 1.

*P13-l6: Sentence starting with 'For the 50-60°SN latitude': please clarify. Why is seasonal variation handled differently in this latitude range?*

The usage of heat fluxes in this latitude range was done following Gebhardt et al. 2014. At these latitudes the seasonal cycle is found to have a strong inter-annual variability that can be insufficiently modeled employing the harmonic terms. Eddy heat fluxes are related to the wave forcing influencing in turn the BDC. The explanation was added to the paper.

*P13-l14: Several studies are mentioned but only one (Park et al., 2017) is cited.*

Yes, we started the sentence with Park et al., avoiding the inconsistency.

*P13-l21: The solar cycle is also used as a proxy in MLR regression of total ozone for trend retrieval in various studies including that from Weber et al. (2018). The solar activity has thus an impact on ozone also in the lower stratosphere. This is worth mentioning.*

The discussion of this proxy has been extended as requested, including the studies of Soukharev and Hood (2006) and of Maycock et al. (2016), investigating the ozone response to the 11-year solar cycle as a function of altitude and latitude from several satellite data sets.

*P17-Fig. 8: Mention for which merged data set are the trends displayed. Trend results should be restricted to the range of validity of the data.*

Thanks, we specified that we are using the anomalies data set and plot the trends up to $\pm\ 70°$ latitude, in agreement with the specified range of validity.

---

## Author Comment (AC2) · 11 Feb 2019

**Replies to Referee #1 on the manuscript 'Retrieval of ozone profiles from OMPS limb scattering observations' by C. Arosio et al.**

We thank the reviewer for the time she/he spent reading the manuscript and constructively commenting on the paper. In the text below, we address the comments from the Referee #1. Referee's comments are shown in italicized font and authors' responses are highlighted in blue.

**1 Short resume** *The construction of long-term ozone profile data records for trend studies has been a lively line of research these past few years. The objective is simple, yet has proven hard to realise: obtain a climate data record which is sufficiently stable (better than a few % per decade) over multiple decades and which ideally covers (most of) the globe at high spatial resolution. Arosio et al. explore two established, complementary merging methods to combine measurements by two dense limb samplers SCIAMACHY (2003-2012) and OMPS-LP (2012-now) using a third limb sensor as transfer standard, Aura MLS (2005-now). Contrary to most earlier efforts by other groups, the authors attempt to preserve longitudinal information. The resulting data record is then analysed for trends over the 2003-2018 period using a widely used regression model. The authors discuss the spatial structure of the trends and they claim that longitudinal patterns are discerned that are indicative of changes in the Brewer-Dobson circulation. The paper concludes by extending the SCIAMACHY-OMPS data record to earlier decades using similar techniques and SAGE II measurements. Zonally averaged trends from the longer record confirm earlier findings for the 1985-1997 and 1998-2018 periods.*

**2 Recommendation**
*This paper fits the scope of AMT and would be suitable for ACP as well, since equal shares of the manuscript are devoted to merging methods and trend analysis results. I would recommend publication as long as the authors are willing to improve the discussion of several topics.*

We thank the reviewer for the positive evaluation, we addressed each comment at our best.

**3 Major comments**
*3.1 Demonstrate that the longitudinal structures are realistic*
*My main criticism on this paper is that there is no substantial proof of the robustness of the reported longitudinal structure of the time series and derived trends. A much more profound discussion is needed about the validity of the longitudinally-resolved results, especially since this is one of the central points of the paper. Constructing a lon-resolved data record is one thing, but the authors need to show that the longitudinal information in the data record is reliable and stable. This should have been the cornerstone of this paper, but it is entirely missing from the paper. As an illustration (p.12, l.2): "A plot of the longitude-resolved drift values is shown in the Supplements, Fig. S1". But no discussion of key results follows: is there longitude structure in the drift field, or not? Another check would be to compare lon-resolved maps of trend results at neighbouring vertical levels to demonstrate their stability in the vertical domain. Once this validation step is over with, you could gain additional confidence by discussing how the derived trend fields compare to what is expected.*

We agree with the reviewer that our discussion of the longitudinally-resolved drift and longitudinal structures observed in the trends was insufficient. We followed the reviewer's suggestions and analyzed in more details the longitudinal structure of the drift with respect to MLS and the vertical consistency of the trend patterns found in Fig. 8.
In the supplements we added the longitudinally-resolved plot of the drift at 41.3 km (Fig. S2, lower panel), which can be compared with Fig. 8, showing ozone trends at the same altitude.

Looking at the drift at this altitude, we see a longitudinal structure: although values are generally non-significant and within ±2%, negative values are found in the [0°, 80°] longitude band, whereas positive drift, yet mostly non-significant, are detected within [100°, 260°] longitude. These patterns do not explain the features found in the trends map, even thought they possibly enhance them. The summary of the results has been added to the paper.

To better evaluate the vertical stability of the longitudinal patterns found in Fig. 8 we added in the paper a second panel to this figure, showing the cross section at 60° N of the trend values. In the Supplements, similar plots at 60° S and in the tropics are provided (Fig. S6). We notice that in the northern hemisphere the positive trends over Canada have statistically significant values over a vertically coherent region of the stratosphere, around 40-45 km; at the same time over the Siberian sector values are statistically non-significant at almost all altitudes. In the southern hemisphere the structure is also vertically consistent without large changes as a function of longitude.

To our knowledge no extensive studies of ozone trends as a function of longitude have been published. However, several studies addressed the longitudinal variations of the BDC, using both satellite measurements and atmospheric models. Kozubeck et al., 2015, found not only a two-core structure of opposite meridional winds in the upper stratosphere at northern mid-latitudes, but also significant positive trends in this structure, studying the last 20 years. Bari et al., 2013, studied the 3D structure of the BDC comparing a general circulation model and MLS observations. The authors found zonal asymmetries in the meridional mass transport, affecting also the ozone and water vapor distribution, especially in the northern middle winter stratosphere. The results of these studies are discussed in the revised manuscript in Sect. 4.2.

*3.2 Elaborate discussion of merging technique*
*The authors present two merging methods and the resulting difference time series with respect to Aura MLS (Figs. 3-4). Unfortunately, they miss the opportunity to discuss merits and weaknesses of each of the methods and in what way one or the other method can correct for specific issues. I feel such a discussion in Sect. 3 would improve the paper a lot. In the end, readers of this paper will be interested in what you recommend as merging approach: plain-debiasing or anomalies? The answer to this naive question may depend on the use case, of course, but this should be part of the discussion. For instance, Fig. 4 shows a discontinuity of in the anomaly-merged time series between 10 N-10 S at 31.5 and 34.8 km. What is the cause of the feature and why is it not present in the plain-debiased time series (Fig. 3)? The trends in the tropics (Fig. 6) are, surprisingly, not very different using both data records. How can this be? On the other hand, p.14, l.14 claims "The general picture in the two panels is very similar, even though trend values in panel (b) are slightly larger". Can this observation be linked to the merging strategy?*

Also the reviewer #2 raised a similar question and the answer to it is not straightforward: as the referee already suggests in the comment, it depends indeed on the use of the data set. The subtraction of the seasonal cycle before merging is generally considered a good way to take into consideration the different geometry of observation of several satellite instruments, when merging them. In our case, since SCIAMACHY and OMPS-LP have a similar sampling and geometry (in terms of scattering angle) we performed also a direct merging, removing just the bias between the two time series, with the advantage of providing a long-term time series directly in terms of number density and preserving the original seasonal cycle. On the other hand, at some latitudes, especially towards the polar region, the differences between the seasonal cycles of the two instruments become important and a direct merging of the two time series in terms of number density may introduce artifacts. The different approaches use the same procedure to compute trends (except for the harmonic terms) and, as we showed in the

paper, do not significantly affect the results of long-term ozone changes. So, in this example of merging two data sets from similar instruments we don't see strong advantages for trend studies using one of the 2 methods. However, depending on the use of the data set, one approach may be more convenient than the other. For example, for data assimilation and model studies, a time series in terms of number density or VMR is more valuable than ozone anomalies, to which a reconstructed seasonal cycle have to be added before the use. We added a short discussion of this issue in the conclusions and extended the description of Fig. 4 in the paper.

The reviewer is right about the discontinuity visible in Fig. 4 in the tropics particularly at 34.8 km. The reason for this feature is that data were plotted without debiasing the anomalies. Indeed, after removing the seasonal cycle from each instrument time series, OMPS-LP anomalies are debiased with respect to SCIAMACHY, using the overlapping period with MLS. This step was missing in the plotted data. The bias is particularly large in the middle tropical stratosphere, where MLS seasonal cycle shows the largest change over the considered period. The plot has been updated and now Fig. 3 and Fig. 4 look, in this respect, much more consistent.

The slightly larger values in the trends using the plain-debiasing approach are most likely related to the different merging strategy, in particular to the way of computing relative trends. In the anomalies approach, the absolute anomalies are divided by the seasonal cycle to obtain relative anomalies, which are directly used to compute ozone trends in % per decade. In the plain-debiased approach, the trends are computed using ozone number density time series and then normalized to the averaged time series at each altitude, latitude and longitude to obtain the values in terms of % per decade. This is now mentioned in the manuscript right after the sentence mentioned by the reviewer (lines 19-22, p.16 of the revised version).

*3.3 Absolute vs relative offset corrections*
*The adopted plain-debiasing method (Eqs. 1-2) removes additive biases but not the multiplicative biases. And vice versa, the adopted anomalies method (Eqs. 3-4) removes multiplicative biases but not the additive biases. Can the authors clarify the statement on p.8, l.8-9: "Through the merging process biases will be subtracted, whereas the discrepancies in the shape of the seasonal cycle are accounted for when calculating anomalies (subtraction of the SC)."?*

The reviewer is right, the additive bias are removed only in the plain-debiasing approach, whereas in the calculation of relative anomalies, the additive bias remains at the denominator. We changed the sentence in the paper accordingly: 'Through the merging process additive biases are subtracted via the plain-debiasing procedure, whereas the multiplicative bias and the discrepancies related to the different shape of the seasonal cycle are accounted for when calculating anomalies.'

We computed, in addition, trends using absolute anomalies, i.e. without dividing by the seasonal cycle as in Eq.4. In this way the additive bias is removed. Trends values computed using absolute anomalies are then divided by the averaged merged ozone time series at each altitude, latitude and longitude. The results are displayed here in Fig. 1 panel (a) and are very similar to the one showed in the paper (Fig. 6 panel (a)), stressing the consistency of the different merging strategies. In particular, we present in panel (b) also the difference map between the zonal trends computed using relative and absolute anomalies: values are extensively within ± 1%.

*3.4 Substantiate claim about stability MLS seasonal cycle*
*p.8, l.3-4: "In addition, we notice that MLS SC may vary within the instrument life time, as shown at 34.8 km in the tropics with change of up to 5–7 % between the two periods." This is quite a bold statement which may worry the users of Aura MLS data. But this claim is not*

[Figure]

[Figure]

Figure 1: In panel (a) zonal trends obtained using absolute merged anomalies over 2003-2018, as a function of altitude and latitude. In panel (b) difference map between trends shown in Fig. 6 panel (a) of the paper, i.e. following the relative anomalies approach, and trends from panel (a) of this picture.

*really substantiated by the authors. Should a reader be really worried about the stability of MLS data, while you mentioned earlier on that it is stable? When looking at Fig. 2 only one panel indicates that MLS 2005-2012 deviates clearly from MLS 2012-2016. I would like to see more proof/discussion if you want to keep the statement that MLS SC varies over its life time.*

We agree with the reviewer that our statement could be confusing for the reader. The seasonal cycle is affected by the atmospheric natural variability and may change depending on the considered period of time. We did not mean to relate the changing MLS seasonal cycle to instrumental problems. We clarified the sentence in the paper, rephrasing it as follows: 'In addition, the natural variability of the atmosphere plays an important role, with the seasonal cycle that naturally evolves with time: we notice, for example, that the seasonal cycle measured by MLS varies between the two considered periods, in particular at 34.8 km in the tropics, where a change of up to 5–7 % occurs.'

*3.5 Collapse of longitude dimension*
*p.7, l.15: "In this paper we only describe the analysis of the longitudinally resolved ozone profile product". If you do not consider the zonally averaged data in this Section, why mention the binning at all? This is confusing as most of the plots in Sect. 3 are latitudinal cross sections. More importantly, the authors do not clarify in what order and how the different dimensions are collapsed from the underlying alt-lat-lon-time resolved data, perhaps because it has no importance but -in that case- it should be stated somewhere. For e.g. Figs. 3 and 4, did you collapse longitude dimension before computing the difference to MLS, or, first compute difference to MLS then average over longitude?*

Several binning possibilities were introduced to show that different products with a higher spatial resolution (in terms of latitude and/or longitude) or a higher temporal resolution are available using data from SCIAMACHY and OMPS-LP. We chose then to consider throughout the paper the following product: monthly mean profiles, spatially gridded every 5° latitude and 20° longitude. However, we did not always show in the manuscript longitudinally resolved results. Indeed, in some cases the longitude-resolved information is not possible to be shown, as it would need too much space (for example for Fig. 3 and 4), in other cases we want to show and compare zonally averaged results from previous studies (for example Fig. 6), or simply we considered as not relevant to show the longitude-resolved plots (for example the correlation

between the merged data set and MLS). We clarified this point in the paper: 'In this paper we consider the longitudinally resolved ozone profile product, i.e. monthly averaged profiles every 5° latitude and 20° longitude. In some cases however we don't show the longitudinally resolved results, either for lack of space or because the zonal averages are directly comparable with previous studies. In this case, the average over longitudes is performed on the level 3 data.'

We thank the referee also for the second point regarding the way zonal mean were calculated. The procedure we followed involves first the computation of ozone zonal mean profiles and then the comparison with MLS data. We included this clarification in the paper before the introduction of Fig. 1.

*3.6 Diurnal variation*
*p.9, l.24-25 reads "Furthermore, at these altitudes diurnal variation of ozone have to be accounted for (Sakazaki et al., 2013), which was not done in this study". This message is repeated on p.15, l.5-6. The correction scheme Eq. 1-2 removes (additive) biases between data records, irrespective of the nature of the bias. Biases due to diurnal variation are part of the total bias. Hence, I infer that diurnal variations are accounted for contrary to what the authors claim. Can the authors respond to this reasoning, and incorporate their answer in the manuscript?*

We see the point of the referee and we generally agree with his/her reasoning: diurnal variations are largely removed when the debiasing is performed. This however holds only if they don't change in time between the two instruments. Analyzing diurnal variations for different months above 45 km, we found not only a variability in shape and absolute values over the year, but also that the difference in ozone between the overpass time of SCIAMACHY and OMPS-LP has a different seasonal cycle with respect to the ozone seasonal cycle at the same altitude and latitude. In addition since above 45 km the solar influence gets more important and complex, we believe that a plain debiasing cannot fully account for this. However we deleted the reference to diurnal variations when describing the zonal trends on pg.15, l.5-6. The Reviewer #2 also raised some concern the about diurnal variation: the reply can be found at p.2 of the answers to the comments.

We included the following sentence in the paper: ' [...] the expected systematic bias between the two instruments is largely removed by the debiasing procedure, even though not completely, because variations with time of this systematic discrepancy may not be accounted for by a 'plain-debiasing"

*3.7 Impact of using ERA-Interim data to convert Aura MLS data?*
*The authors mention that the Aura MLS data record is stable (p.6, l.18-19). However, it is not clear from the paper whether this holds for converted Aura MLS data as well. Please elaborate on how the ERA-Interim data may impact the converted Aura MLS data. Can it induce the change in seasonal cycle reported on p.7, l.6-7? Can it lead to the drift above 50 km reported in p.9, l.22-23?*

Also the referee #2 raised some concern about MLS conversion into number density: we performed the conversion of MLS data using pressure profiles from MERRA-2 instead of ECMWF ERA-Interim. The results are shown in the appendix of the paper and demonstrate that the effect on the ozone trends is negligible. However, relative differences between MLS profiles converted using the two different reanalysis were found to reach 3-5% above 55 km.

We don't expect this conversion to cause the change over time of the MLS seasonal cycle reported in the paper, which is mostly related to the natural variability, since it is particularly evident only in the middle tropical stratosphere.

The observed drifts in Figs. 3 and 4 above 50 km are too large to be caused by the MLS conversion. We show in Fig. 2 (see below) the drift of the difference between MLS converted using ECMWF ERA-Interim and using MERRA. The drift is in terms of % per decade and computed over the 2005-2016 period. We can notice that above 50 km the drift is around 1–1.5%, smaller than the short term drift found in Figs. 3 and 4. In addition, we did not find any change of sign in this drift over SCIAMACHY and OMPS time, as displayed in the picture in the paper.

[Figure]

Figure 2: Drift of differences between MLS converted using ECMWF ERA-Interim and using MERRA, as a function of altitude and latitude. The drift is in terms of % per decade, computed over the 2005-2016 period.

*3.8 Correlation between solar and trend term*
*The MLR regression model contains a term for the 11 year solar cycle and a linear trend term (p.12, Eq.6). The analysis period (2003-2018) contains one and a half solar cycle, which triggers the question as to how independent the two said low-frequency terms are. Can the authors elaborate on this? Could the change in derived trend for different starting times (p.15, l.7-10) be related to interference between the solar and trend term, this is exactly the region where solar influence should be large. This concern may even be more important for the results shown in Figs. 7 and 9 where even shorter periods are regressed. Perhaps in these cases the non-trend terms were regressed over the entire time period?*

The reviewer is right about the issue and we are aware of the problem. That's why we specified in the paper to take the trends over short time periods with caution. We are also aware that more robust results should come from a record covering 2 full sun cycles, but for that we have to wait some more years.
As suggested by the reviewer at this point and in a couple of minor comments, we also tried a different way to regress the trends over the shorter periods (2004–2011 and 2012–2018). First we fit all the proxies excluding the linear terms over the longer time period (2003-2018) and then we perform a linear fit for shorter periods using the residuals after the first step (differences between time series and fit). As an example we show here in Fig. 3 the comparison between the zonal trends over 2004-2011, computed using both methods: in panel (a) the same as the plot presented in the paper and in panel (b) using the wider time range to fit all the non-trend terms. We put this figure in the Supplements (Fig. S7). As we notice, the differences are rather small, even though the bipolar pattern found in the tropics and southern mid-latitudes is less pronounced using the second strategy. However, this method can be applied for the 2 considered sub-periods but we don't see how it could be applied for the 2003-2018 SCIAMACHY-OMPS merged data sets, in which case there are no measurements to be fitted before 2003.

[Figure]

Figure 3: In panel (a) the trends are computed fitting all the terms over the 2004-2011 period (as done in the paper), in panel (b) all the non-trend terms were fitted over the 2003-2018 period. In panel (c), we present the differences of zonal trends over the period 2003-2018, with and without considering the solar proxy.

In order to evaluate the magnitude of the effects of the solar proxy on long-term ozone variations, we computed trends over 2003–2018 without considering the solar proxy in the fit: panel (c) of this figure, shows the difference between the results of this calculation and panel (b) of Fig. 6 in the paper. The differences are within ± 1% at most altitudes and latitudes, so that we expect that the interference between the solar and trend term pointed out by the reviewer is smaller than this threshold.

**4 Minor comments**
*p.1, l.11: Be specific about what you mean with "remarkable variability".*

We find it difficult to be specific in the abstract, however we added: 'with variations of up to 3–5 % per decade at altitudes around 40 km.'

*p.2: Very nice and concise overview of ozone-related processes.*

Thanks.

*p.3, l.19: Identify "MLS" as "Aura MLS" here and throughout the rest of the paper. You don't want to confuse with the first MLS instrument which was flown in the 1990s-2000s on the UARS satellite.*

Thanks for the observation, we specified that we mean the MLS instrument onboard the Aura

satellite to avoid confusion the first time we introduce the acronym, but we kept the use of MLS to indicate Aura MLS, in the rest of the paper.

*p.3, l.19-21: You should introduce SAGE II over here, instead of two instruments (ACE FTS and SAGE III/ISS) that are not mentioned in the rest of the manuscript.*

We agree with the reviewer to mention at this point the SAGE II instrument but we also left the sentence about ACE FTS, MAESTRO and SAGE III as currently operating solar occultation instruments.

*p.3, l.25: Please rephrase. Harris et al (2015) did not merge these data sets, but use them to derive trends.*

Thanks for this comment, also the Reviewer #2 highlighted this point: we rephrased as: '... the authors considered several existing merged satellite data sets and examined separately the time spans before and after the peak in ODSs concentration at the end of '90s. The authors combined trends from the different data sets and ...'

*p.3, l.34: Remove "applying a multilinear regression analysis". This information is evident and not different from the other analyses you refer to.*

True, we deleted this information.

*p.4, l.1: Vague statement "Ball et al. (2018) applied a method independent from the ozone turnaround point". The subsequent clause "showed for the first time some evidence of a negative trend in lower stratospheric ozone" seems to imply that the different regression method is leading to this discovery. I am not sure that is what Ball et al. claimed.*

We improved the sentence as follows: 'The authors analyzed a longer period of time, together with improved merged time series and considered the lower stratospheric column instead of the ozone profile. With these adjustments, they showed for the first time ...'

*p.4, l.4-5: Hanging statement: "This analysis has recently been challenged by Chipperfield et al. (2018)". In what way?*

We added, 'who showed that the apparent downward trend in the lower stratosphere (ending in 2017) is a result of longer term variability in atmospheric dynamics.'

*p.4, l.6: Clarify what a "pointing drift" is. A general reader will not have a clue what pointing means in this context. Consider vertical pointing, altitude registration, ...*

Thanks, we replaced with 'altitude registration', which is more clear: '..., after OSIRIS data were corrected for a drift in the tangent altitude registration of the instrument.'

*p.4, l.15: I am not sure LOTUS is "homogenizing" the merging procedures, please double check this with one of the LOTUS participants.*

We replaced it with '... studying robust methods to merge data sets ...'.

*p.5, l.12: "performing measurements at 3 viewing angles, which differ horizontally by 4.25 deg.*

Yes, we reformulated the sentence as suggested by the reviewer.

*p.5, Tab. 1: Extend this table to SAGE II and Aura MLS.*

We added the information for MLS and for SAGE II.

*p.5, Tab. 1: Add level 2 versions in this table as well. This will make life easier for readers in 5 years from now.*

Yes, it is indeed a good idea.

*p.5, Tab. 1: I advise to show the analysis time period for both instruments. Right now, different information is conveyed in "data time series": SCIA (full mission period) and OMPS (analysis time period). Please use one or the other, but do not mix up.*

Thanks, we restricted to the time periods used in our analysis.

*p.5, Tab. 1: Align values of spectral coverage and spectral resolution with what is in the main text.*

Yes, for SCIAMACHY there was a discrepancy.

*p.5, l.20: Add the version of SCIATRAN.*

We included the respective versions: v3 for SCIAMACHY and v4 for OMPS.

*p.6, l.1: Add the version of the SCIAMACHY L1 data, as was done for OMPS-LP.*

We included the following sentence: 'In particular, v8 L1 SCIAMACHY and v2.5 L1 OMPS-LP data were processed.'

*p.6, l.9: Clarify "pointing knowledge issues", see also my earlier comment. E.g. "[...] when the issues related to the vertical pointing of the instrument, currently under [...]"*

The sentence has been modified as follows: 'data from the lateral slits are planned to be used when the issues related to the tangent altitude registration of the instrument, currently under investigation by NASA, are solved.'

*p.6, l.9: Refer to Moy, AMT 2017.*

Done

*p.6, l.13: Replace "scientific measurements" by a better description or simply drop "scientific".*

We replaced it with 'atmospheric observations'.

*p.6, l.18: You mention Hubert et al. (2016) for Aura MLS drift. But any trend paper should refer to published drift results for all instruments involved. I.e. add those for SCIAMACHY (Rahpoe-2015, Hubert-2016, LOTUS-2018, ...?) and OMPS-LP (Kramarova-2018) as well, un-*

*less the L1-L2 versions have changed sufficiently to question the validity of those values.*

Both Raphoe 2015 and Hubert 2016 refer to the old version of SCIAMACHY L2 data, while the LOTUS second report still has to be published. The only work to our knowledge which addressed the drift with respect to other satellite data sets of SCIAMACHY v3.5 ozone, is Sofieva et al., 2017. The authors stated that evaluating and inter-comparing the anomalies of the considered instruments, among which SCIAMACHY, they did not find statistically significant drifts with respect to the median anomaly. This was summarized in the paper. We also added a reference to Kramarova et al., 2018, as suggested, addressing OMPS-LP drift with respect to MLS and OSIRIS.

*p.6, l.18: Find a better phrasing for "For technical reasons" as it suggests an instrument malfunction. The observations by SAGE II are sparse due to the chosen measurement geometry and is unrelated to the instrument itself.*

We agree with the reviewer: we changed the sentence as 'Due to the occultation viewing geometry, ...'.

*p.6, l.31: "Aura MLS", see earlier comment.*

We specified at the beginning that with MLS we mean Aura MLS.

*p.6, l.33: "taking only the latitude covered daily by OMPS-LP". You lost me here, what latitudes are not covered by OMPS? Please clarify what you mean in the main text. And why does this resolve an inconsistency? SCIAMACHY measurements are also made during daytime.*

Yes, here we need to be more clear: for every day of the year, we considered MLS measurements within the latitude extremes covered by OMPS-LP during that day. They do not coincide with MLS profiles flagged as day-time observations in L2 data. We reformulated the sentence as: 'For each day, we take only MLS measurements which are made within the latitude range covered by OMPS-LP and SCIAMACHY.'

*p.7, l.2: "Aura MLS", see earlier comment. Please incorporate this comment in the rest of the manuscript.*

As mentioned, we specified at the beginning that with MLS we mean Aura MLS.

*p.7, l.3: Add a motivation for not using the 2002 SCIAMACHY data.*

Some studies, like Sofieva et al. 2017, reported anomalous values at the beginning of SCIAMACHY mission with respect to other satellite observations. We specified it in the paper.

*p.7, l.11-12: Figure 1 does not confirm the statement "In both cases we find about 100 profiles on average in each bin." for SCIAMACHY. Each monthly 5deg zonal bin has 1000 profiles, which translates to 56 (=1000/18) profiles per bin, not 100. Did I misunderstand? If not, please change the misleading statement.*

With 'about 100 profiles' we meant the order of magnitude, aware that for SCIAMACHY the number is lower than for OMPS-LP and that the available observations depend on latitude and time. Instead of 'about 100', we wrote 'on average 50–100'.

*p.7, l.14: Clarify what interpolation method was used.*

Done, we mentioned that we used a linear interpolation.

*p.7, Fig.1: Add 5° at the end of the caption: "[...] in each 5° zonal monthly bin [...]".*

Done

*p.8, Fig.2: Is the SCIAMACHY time period identical to that of Aura MLS (2005-2012)? Please add the time period for all four lines in the legend, not just for Aura MLS.*

Yes, seasonal cycle for OMPS-LP and SCIAMACHY are computed using the same period as for MLS. We included the time periods for all the instruments in the legend of Fig. 2 and updated the description in the paper.

*p.8, l.5-6: The phrasing is not clear whether the time period was only adapted for MLS. In other words, did you compare 2005-2012 for both SCIAMACHY and MLS, and 2012-2016 for both OMPS-LP and MLS? See previous comment.*

See previous comment

*p.9, l.14: Add a short phrase that the unit of the plain-debiasing data set is ozone number density.*

We added this information in the same sentence: 'The merging is then achieved by concatenating the two data sets, in terms of ozone number density, ...'

*p.9, l.16: "[...] differences between the merged data set [...]". What merged data set? The zonal one? The longitudinally resolved one?*

In the sentence before this one we spoke about the merging of the 2 time series, so we are here directly referring to the plain-debiased longitudinally resolved data set. The picture shows however zonally averaged differences: this has been done for technical reason, it is already a pretty busy figure and we did not find an easy way to show also the longitudinal dimension, which in any case would not add much information. We explained this in the paper when we introduce the binning of the data sets. The collapse of the longitudinal dimension is done again on the Level 3 data: MLS, OMPS and SCIAMACHY data sets are firstly zonally averaged and then the processing (computing, bias, anomalies etc...) is performed.

*p.9, l.16-18: What is the sign of the relative difference? (SCIAOMPS - MLS) / MLS or the other way around?*

It is (SCIA or OMPS - MLS) / MLS. We added this information in the paper and it is valid for the whole paper.

*p.9, l.28-29: Are the larger relative difference values at 15 km truly due to lower data quality or due to the smaller number densities in the UTLS region?*

We think it is for both reasons, for sure the smaller number density in the UTLS region especially in the tropics leads to larger relative differences. We added at p.12, l.19-20 of the revised version that: '... low values of ozone number density, especially in the tropics, amplify the relative differences.'

*p.9, l.30-31: Replace by "[...] deseasonalized relative anomalies from [...]" to clarify that you are not working with absolute anomalies.*

Done

*p.9, l.30-31: What is the motivation behind debiasing the deseasonalized relative anomalies? By computing the anomaly any multiplicative biases are removed by definition.*

SCIAMACHY, MLS and OMPS-LP time series are deseasonalized over different periods and have then a zero mean values over their respective periods. So that an additional debiasing using MLS is required, which consists in bringing OMPS-LP anomalies to the average values of MLS anomalies over 2012–2016 and then to SCIAMACHY anomalies over 2005—2012.

*p.9, l.31: Replace by "[...] month of the year, m, the (relative) anomalies, [...]".*

Done

*p.10, Fig. 3 and p.11, Fig. 4: Add in the colour scale the exact sign of the difference: (merged - MLS) / MLS or (MLS - merged) / merged ?*

We referred in the caption of Figs. 3 and 4 to Eqs. 3 and 7, which explicitly express how the differences were computed.

*p.10, Fig. 3 and p.11, Fig. 4: Add in the caption that MLS data has been offset to SCIA prior to the comparison.*

Done, using 'before the comparison' instead of 'prior to the comparison'.

*p.10, l.4: Eq. 5 is not really used in the rest of the paper (p.12, l.7-8). I would therefore suggest to drop it, also because (a) you do not explain how the uncertainty for the plain-debiasing data is computed and (b) there is no term for the uncertainty in the seasonal cycle.*

We agree with the referee, the equation was removed.

*p.10, l.9: Replace by "Figure 4 shows the absolute differences [...]" to clarify that these are absolute differences of relative anomalies.*

Done

*p.11, l.1: "whereas below 20 km the pattern becomes rather chaotic".*

The comment is not complete and thus cannot be addressed.

*p.11, Fig.4: Larger differences are found around 35 km in 10S-10N during the OMPS-LP period. What is the cause of this? Does the different MLS SC in the two periods play a role?*

We updated this picture, as mentioned in the main comments: now the differences are smaller, therefore we dropped the sentence.

*p.11, l.4-5: "The drift is computed as the linear change of the differences between the merged time series and MLS data [...]". Are these relative differences for plain-biased merged data and absolute differences of anomaly-merged data? Clarify this in the text. And add the unit of the drift: % per decade/year/... .*

We specified that the differences are either relative Eq. 3 or absolute Eq. 7 for the 'plain-debiased' data set and anomalies respectively'. The unit of the drift was added.

*p.12, l.1-2: "Very similar results for the drift are obtained using anomalies time series". The timeseries in Figs. 3-4 look fairly different in places, and I am surprised the drift results are very similar for the anomaly time series. This plot has to be in the main paper, also since it may be the basis of an interesting discussion on what merging technique led to most stable results for this particular case. (See also one of my major comments.)*

Since the plots of the drifts computed using the two strategies are very similar we decided to keep the other plot in the Supplements (Fig. S1) and we refer to it in the paper.

*p.12, Fig. 5: Add in the colour scale the full unit (% per decade/year/...) and the exact sign of the difference: (merged - MLS) / MLS or (MLS - merged) / merged ?*

We added the full unit and we refer to the equation expressing the sign of the difference.

*p.12, l.6: What are the units of O3 in Eq. 6? The plain-debiased time series are in molec $cm^{-3}$, the anomaly-merged time series are in %?*

We specified the units also at this point in the paper.

*p.12, l.14-15: The phrase "The t-th row of the X matrix contains the values of the fit terms for the selected t." does not add information. It could easily be dropped.*

We see the point of the referee, we dropped the sentence.

*p.13, l.1: The equivalence of the $2\sigma$ rule to 95% confidence level is introductory statistics, hence the reference to (Tiao et al., 1990) is not needed.*

Done, we agree with the reviewer.

*p.13, l.3: How are the plain-debiased time series in molec $cm^{-3}$ regressed to obtain % per decade?*

The time series are regressed in terms of *molec $cm^{-3}$*, then the obtained trend values are divided by the mean ozone over the time series at each altitude-lat-lon bin. We added this information in the paper.

*p.13, l.6-9: Do I understand you correctly that the EHF term is used instead of the harmonic terms, below 25 km and only for the 50-60° N band? Why not for 50-60° S as well, or at other latitudes? Please add that this modified regression model is not applied to the analysis of*

*anomaly-merged time series.*

We refer here to the work of Gebhardt et al. 2014, where it was found that at these latitudes in the northern hemisphere the ozone annual cycle has a larger amplitude with respect to the southern hemisphere, due to the stronger wave activity at northern mid-latitude, which influence the ozone distribution in the lower stratosphere in this region. For the same reason, the annual cycle is also characterized by a strong interannual variability which can lead to an insufficient modeling of the seasonal cycle if considering simple harmonic terms. That's why we used EHF integrated starting from October of each year. The explanation has been improved in the paper.
In addition, EHF are used to regress both the ozone number density time series and anomalies.

*p.13, l.23: Please cite more recent work, at least Maycock et al. (2016), perhaps others as well (Ball et al., 2016; Damadeo et al., 2018; ...).*

We agree with the referee, the paragraph was revised citing the studies of Soukharev and Hood (2006) and of Maycock et al. (2016).

*p.14, l.1: Add the source of the El Nino 3.4 index data, as you did for the other proxy data sets.*

Done: 'The data time series is available at:
$http : //www.esrl.noaa.gov/psd/gcos\_wgsp/Timeseries/Nino34/.$'

*p.14, l.6: Add that N34 represents the El Nino 3.4 index anomaly data.*

We think that this is already addressed, since we say on p.6 l.1-2 (p.16, l.4-5 of the new version) that '... [El Nino 3.4 index] is based on sea surface temperature anomalies averaged from 5° S–5° N and 170°–120° W.'

*p.14, l.11-12 and Fig. 6: Are the trends in Fig. 6 regressed directly from the zonally averaged merged time series, or are these trend results regressed from lat-lon resolved merged time series then averaged over the latitude bands? In the first case, this contradicts an earlier statement that only analysis of lon-resolved data would be described (p.7, l.15). In the latter case, how do you propagate the uncertainties?*

The reviewer is right. Our initial statement was probably not clear: as already addressed in the major comment 3.5, we refer to the lon-resolved data set meaning that throughout the paper we considered monthly profiles binned every 5° latitude and 20° longitude. However some of the presented plots show zonally averaged results, due to lack of space to show the lon-resolved fields or for the relevance of the zonal averages. In this case, the zonally averaged L3 time series were considered and regressed to compute the trends. The explanation was added at the beginning of Sect. 3.

*p.15, l.2: You may consider adding the LOTUS report and replace the WMO reference in this phrase (WMO, 2018).*

We updated the WMO reference, while LOTUS report is not yet published to our knowledge.

*p.15, l.9: Perhaps the cause is not instrumental, but related to the interference of the solar and*

*trend term? See my major comment above.*

This is also a possibility; we addressed this issue at the major comment 3.8.

*p.15, l.35: Are all terms (seasonal, QBO, solar, ENSO, ...) constrained by the 2003-2011 period or just the linear trend term? This shorter period potentially makes the interference between solar and trend terms even larger. Have you looked into this? The trend results may be more stable/robust when you constrain all non-trend terms (especially solar) to the larger 2003-2018 period.*

The plots presented in the paper show trends computed fitting all the proxies in each respective period, in this case 2004-2011. We addressed this issue replying to the point 3.8 of the referee's Major comments.

*p.16, Fig. 7: Add complete unit to y-axis label: molec $cm^{-3}$.*

We added the unit to the caption.

*p.16, Fig. 7: Adding the anomaly-merged time series and fits would make a fine illustration of how the merging strategy can overcome some of the issues in the data sets as mentioned e.g. in p.16, l.4-7.*

We realize that the sentence, which the reviewer refers to, may be misinterpreted: we do not explicitly mean here that the the plain-debiasing approach is better, in this regard, with respect to the anomalies strategy. We only meant that at these altitudes and latitudes (tropics, mid-stratosphere), due to the change in time of the MLS seasonal cycle already pointed out, MLS anomalies are sensitive to the period over which the seasonal cycle is computed. As a consequence, also the merged SCIAMACHY/OMPS anomalies data set is affected by MLS deseasonalization. This is not a disadvantage of the anomalies approach by itself but a consequence of the merging of two data sets without an extensive common period.
In order to address the reviewer's request, i.e. add the anomaly-merged time series + fits, another self-standing plot would be needed, since the vertical scale is different. We think that this issue is too technical to be fully discussed in the paper.

*p.16, l.1: There is the switch to OMPS-LP in 2012. Can this be a viable alternative explanation to the "discontinuity"? The fits themselves will, in addition, likely be impacted by the solar-trend interference as well.*

The change in the instrument plays most probably a role, but not to the extend of explaining the jump. This would be the case only if both instruments had a drift of different sign, which has not been so far identified. In addition, we are currently comparing our results with a run of the TOMCAT CTM and we found similar results between 30 and 35 km in the tropics, which gives us more solidity for such a conclusion. Anyway, we added in the paper that: 'In addition, the switch between SCIAMACHY and OMPS-LP time series and the interference between the solar proxy and the trend-terms may enhance this discontinuity in the long-term changes.'

*p.16, l.11-12: Please substantiate why the longitudinal trend results are reliable? A figure like Fig. 8 for neighbouring levels z=41 and 44 km will help to demonstrate the stability of the results in the vertical domain, especially in the US where trends are mostly significant (also in other studies).*

At this aim, we included in Fig. 8 a panel (b) illustrating the vertical cross section of the trends as a function of longitude at 60° N. This shows how the positive significant values over the Canadian sector are vertically consistent between 38 and 45 km.

*p.17, Fig. 8: Add to the caption what merged data set was used: plain-debiased or anomaly-based?*

Done

*p.17, Fig. 8: Remove the results in the latitude-range that you mentioned earlier on was not reliable (60° for plain-debiased and 70 for anomaly-based).*

We used the anomalies data set, so we keep the results up to 70°.

*p.17, l.10-11: Move this discussion to previous paragraph and elaborate on how stable results are in vertical domain.*

We left this sentence here, because the study of Kozubeck et al. refers to the upper stratosphere. However we added here a short description of the plots in the Supplements.

*p.17, l.15-16: Motivate why you use the anomaly approach.*

This was done to take into consideration the different geometry and sampling of the three instruments (particularly, the low density of SAGE II measurements). We included this in the paper.

*p.18, l.6-7: Add brief explanation why the harmonic terms are not included (deseasonalized anomaly time series).*

As done for the SCIAMACHY-OMPS merged anomalies, we don't need the harmonics here indeed because the seasonal cycle is naturally subtracted when calculating anomalies.

*p.18, l.6-7: Slightly confusing, since the trend model is very different from that in previous section. Please clarify whether it is an independent trend (ILT) or a piece-wise trend (PWLT).*

The same trend model is applied to the merged SCIAMACHY-OMPS anomalies data set and to both individual data sets. We specified in the paper that independent trends over the two periods have been computed.

*p.18, l.13: Add correct unit : "[...] about -2% per decade is detected [...]".*

Done

*p.18, l.13: As asked before (p.15, l.35), what time period was used to constrain the non-trend terms? And how robust are -especially- the 2012-2018 trend results given the low frequency of the 11 year solar cycle proxy?*

This point was already addressed in the reply to the referee's major comment 3.8.

*p.18, l.19: You mention 2010-2018 here, while the figure caption says 2012-2018. Which one*

*is correct?*

Thanks, it is 2012-2018, the text was wrong.

*p.18, l.32: What do you mean with "up to" the polar regions?*

'Including' is indeed better.

*p.19, Fig. 9: You mention 2012-2018 in the caption, while the main body text says 2010-2018. Which one is correct?*

Same as two comments before.

*p.19, Fig. 9: Add the time period to each panel, in addition to (a), (b), ...*

We agree, it improves the readability.

*p.19, l.6-7: Strong claim that needs demonstration: how reliable is the observed lon-resolved structure?*

We softened it by saying 'This is an indication of a possible change in the BDC as a function of longitudes in the northern hemisphere'.

*Supplement, Fig. S1: Add in the caption which merged time series are shown: plain-debiased or anomaly?*

Done

*Supplement, Figs. S1 and S2: Add sign and correct unit (% per decade?) to colour scale or in the caption.*

Done

*Supplement, Figs. S1 and S2: Each subpanel represents one longitude-bin, all together they convey information about longitude structure of drift of SCIA OMPS wrt MLS. However, the longitude structure would be much more obvious if you would have shown one latitude-bin per subpanel, and then plot drift vs altitude vs longitude. Can you add this to the supplement?*

It is a good suggestion, we inserted in the Supplements a panel with the drift as a function of latitude and longitude at 41.3 km. In addition cross sections of trends, i.e. altitude vs longitude trends, have been added to the paper in Fig. 8 and in the Supplements Fig. S6.

*Supplement, Figs. S3 and S4: Add to the caption what merged data set was used: plain-debiased or anomaly-based?*

Done, these plots are now Figs. S4 and S5.

*Supplement, Figs. S3 and S4: Remove lat-range with data that you claimed earlier in the paper are unreliable (poleward of 60° or 70° latitude).*

The plots are now up to 70° latitude.

**5 Technical corrections**

*p.1, l.4-5: Replace by "[...] is performed by the processor of the University [...]"*

We reformulated the sentence as follows: 'The retrieval of ozone profiles from SCIAMACHY and OMPS-LP is performed using an inversion algorithm developed at the University of Bremen.'

*p.1, l.10: Replace "high" horizontal sampling by "dense" horizontal sampling.*

Done

*p.1, l.24: Remove either "important" or "key". Important implies key and vice versa.*

We removed that part of the sentence, leaving: 'The continuous monitoring of the stratospheric ozone layer is required to assess the impact of anthropogenic and natural processes.'

*p.2, l.7: Replace by "during the 1990s".*

This sentence was changed: 'The adoption of the Montreal Protocol and its amendments regulated the industrial production of chlorine and bromine compounds: in particular, the London amendment in 1990 called for a complete phase out of CFCs production by the year 2000, leading to a decrease of their concentration in the stratosphere starting from the end of the 20th century.'

*p.2, l.10: As non-native speaker I expected "GHGs such as", but perhaps "such" is not needed.*

We added 'such'.

*p.3, l.27: Add a "," in "[...] before 1998, and a positive trend of [...]".*

Done

*p.3, l.29: Replace by "[...] with uncertainty estimates. [...]".*

Done

*p.3, l.30: Add "-", replace by "[...] satellite and ground-based data sets [...]".*

Done

*p.3, l.35: Replace by "[...] significant trends [...]".*

Done

*p.3, l.35: Replace by "[...] in the upper stratosphere at mid-latitudes [...]".*

Done

*p.5, l.4: Replace "[...] in-flight direction [...]" by "in flight direction" or "in the direction of flight".*

We used 'in flight direction'.

*p.5, l.13: Remove the first "charged" in "charged charged-coupled device".*

Done, thanks.

*p.5, l.22: Replace "application of SCIAMACHY retrieval scheme" by "application of SCIA-MACHY's retrieval scheme".*

Done

*p.6, l.3: Replace "we take into account in addition" by "we also take into account".*

Done

*p.7, l.23: Replace by "[...] the SC of all single instrument data sets [...]".*

Done

*p.8, l.3: Replace by "[...] the three ozone profile data records in number density [...]".*

Done

*p.13, l.16: Replace by "[...] and the in-phase at mid-latitudes [...]".*

Done

*p.13, l.32: Drop "a" in "[...] leading to longitudinally dependent modifications of ozone [...".*

Done

*p.14, l.17: Replace by "Bourassa et al. (2018)".*

Done

*p.15, l.11: Replace "detected" by "found" or "observed". In my view, "detected" implies that the result is significant which is not the case.*

We replaced it with 'observed'.

*p.15, l.33-34: Remove newline after "[...] panel (a).".*

Done

*p.17, l.13: Add "s" to "[...] SAGE II occultation observations [...]".*

Done

*p.18, l.4-5: Replace by "[...] the mean SAGE II latitude plus or minus its standard deviation [...]".*

Done

*p.18, l.7: Replace by "[...] 60° latitude [...]".*

We removed the ±.

*p.18, l.29: Replace by "[...] is first removed [...]".*

Done

*p.18, l.30: Remove "one" from "[...] MLS one [...]".*

Done

*p.18, l.32: Replace by "[...] with respect to the MLS time series [...]".*

Done

*p.19, caption: Replace by "[...] and in panel (d) over 2012–2018 [...]".*

Done

*p.19, l.4: Replace by "[...] from 2003 until early 2018 [...]".*

Done

*p.19, l.5: Replace "detected" by "found" or "observed". In my view, "detected" implies that the result is significant which is not the case.*

We replaced it with 'found'.

*p.19, l.10: Replace by "[...] has vanished when adding [...]".*

Done

---

## Author Response (AR2)

**Replies to Referee #1 on the manuscript 'Merging of ozone profiles from SCIA-MACHY, OMPS and SAGE II observations to study stratospheric ozone changes' by C. Arosio et al.**

We thank the Referee #1 for the time she/he spent reading the manuscript and checking the revision. In the text below, referee's comments are shown in italicized font and authors' responses are highlighted in blue.

*Thanks for the revision. I have only two technical remarks...*
*p.8, l.21-22: replace by "In this case, the average over longitudes is performed on the level 3 data prior to any further computations (e.g., trends, differences, ...)."*

We followed the suggestion replacing this sentence in the manuscript.

*LOTUS Report was recently published as "SPARC/IO3C/GAW, 2019: SPARC/IO3C/GAW report on Long-term Ozone Trends and Uncertainties in the Stratosphere. I. Petropavlovskikh, S. Godin-Beekmann, D. Hubert, R. Damadeo, B. Hassler, V. Sofieva (Eds.), SPARC Report No. 9, WCRP-17/2018, GAW Report No. 241, doi:10.17874/f899e57a20b, available at http://www.sparc-climate.org/publications/sparc-reports."*

*Add this reference "SPARC/IO3C/GAW (2019)" to - end of p.4, l.28-31 - p.16, l.25-26 : next to WMO (2019).*

We agree with the reviewer: we added this reference in both the indicated lines.

**Replies to Referee #3 on the manuscript 'Merging of ozone profiles from SCIA-MACHY, OMPS and SAGE II observations to study stratospheric ozone changes' by C. Arosio et al.**

We thank the Referee #3 for the time she/he spent reading the manuscript and checking the revision. In the text below, referee's comments are shown in italicized font and authors' responses are highlighted in blue.

*I have read the Revised Submission of the paper "Merging of ozone profiles from SCIAMACHY, OMPS and SAGE II observations to study stratospheric ozone changes" by Arosio et al., as well as the two Referees Reports and the Author's Reply. In my opinion, the Authors have very effectively tackled the Major, Minor and Technical Comments of the two Referees; all points have been taken into consideration and the manuscript, which was already very well written and useful in its scientific content, has been further improved.*
*For this reason, I recommend to publish the paper as is, with only a possible further correction: In P3, L3-5, the statement "From monthly up to decadal time scale, ozone concentration is also influenced by many well known phenomena such as the 11-year solar activity cycle and solar proton events, the Quasi-Biennial Oscilation (QBO), El Niño Southern oscillation (ENSO), and volcanic eruptions" maybe calls for a couple of references.*

We thanks the reviewer for the positive response. We agree with this suggestion: we added four references at this point of the paper related to the solar activity, QBO, ENSO and volcanic eruptions.

**Replies to Editor comments on the manuscript 'Merging of ozone profiles from SCIAMACHY, OMPS and SAGE II observations to study stratospheric ozone changes' by C. Arosio et al.**

We thank the Editor for the time she/he spent reading the manuscript and providing the minor corrections. In the text below, editor's comments are shown in italicized font and authors' responses are highlighted in blue.

*p1 l11 write 'to study ozone changes in the 20-50 km range over the 2003-2018 period.'*

Done

*p1 l15/16 I think you mean "which is attributed to both …. and the decrease …."*

Yes, the sentence was accordingly modified.

*p1 l18/19 please add dates behind "SCIAMACHY period". It doesn't seem to have been defined previously*

The period (2002-2012) was added.

*p2 l27 (Portmann et al. 2012) → Portmann et al. (2012)*

Corrected

*p3 l2, consider changing: time scale → time scales*

Done

*p3 l6 ozone number density field ?*

We replaced 'ozone field' with 'ozone distribution'.

*p3 l21 before end of the ENVISAT mission → before the end of the ENVISAT mission*

Done

*p3 l23 Stratospheric ozone profile is → Stratospheric ozone profiles are*

Done

*p6 l4 which spectroscopic databases have been used ? Citation of the data base seems to be missing.*

Thanks, we added two references at this point.

*p7 l4 → with discrepancies in the tropics above 22 km generally below 5%*

Done

*p8 l1 Consider rephrasing: The SCIAMACHY data set of this study covers the period from ...*

We rephrased as 'The SCIAMACHY data set is considered in this study starting from ...'

*p10+ eqs 1-3,7,13 the authors should consider using common symbols for lon et lat, eg $\phi$, and $\theta$ ....*

We replaced 'lat' with $\phi$, 'lon' with $\theta$ and 'alt' with z.

*p14 eq(8) check bold faces in the last line of the equation 8 and define X*

The matrix $\mathbf{X}$ has been defined. Now $O_3$, $\beta$ and $N$ are in vector notation, whereas $\mathbf{X}$ is in matrix notation.

*p15 l8 As a consequences $\rightarrow$ As a consequence*

Done

*p15 l14 in the ozone changes $\rightarrow$ in ozone changes*

Done

*p15 l16 opposite phase in the tropics and the in-phase at $\rightarrow$ opposite phases in the tropics and being in phase at*

Correction implemented.

*p15 l22 25 years period $\rightarrow$ 25 year period (units don't take the plural s)*

Done

*p15 l24 11 years $\rightarrow$ 11 year*

Done

*p15 l27 a reduced variations $\rightarrow$ reduced variations*

Done

*p15 l27 (overtilde(1)) $\rightarrow$ ( 1%)*

Done

*p15 l32 The solar proxy is applied to all latitudes and altitudes, given by: $\rightarrow$ The solar proxy applied to all latitudes and altitudes is given by:*

Done

*p16 l12 within 10 iteration $\rightarrow$ within 10 iterations*

Done

*p16 l16 In panel (a) ...., The sentence is incomplete, verb is missing*

The dot has been replaced by : joining the sentences, so that no verb is needed.

*p16 l27/28 use correct spectroscopic symbols for ozone species: ground state O, use $O(^3P)$, ground state $O_2$ is $O_2(^3\Sigma_g^-)$. Reaction with singlet O2 is not significant, use 'ground state molecular oxygen' instead of 'ground molecular oxygen'*

Thanks for the corrections.

*p18 l6 Model studies for the 2004-2018 period are ongoing. Why 2004-2018 and not 2003-2018 ?*

It was a mistake, we put 2003-2018.

*p19 l13 a similar distributions → similar distributions*

Done

[revised manuscript text omitted]
}(\underset{\sim}{lat}\phi, \underline{lon}\theta, z) = mean(SCIAMACHY_{2005-2012}(\underset{\sim}{lat}\phi, \underline{lon}\theta, z)) - mean(MLS_{2005-2012}(\underset{\sim}{lat}\phi, \underline{lon}\theta, z))$$

(1)

$$BIAS_{OMPS}(\underset{\sim}{lat}\phi, \underline{lon}\theta, z) = mean(OMPS_{2012-2016}(\underset{\sim}{lat}\phi, \underline{lon}\theta, z)) - mean(MLS_{2012-2016}(\underset{\sim}{lat}\phi, \underline{lon}\theta, z))$$

In these and following equations, ozone profiles from each instrument are considered as binned monthly averages, interpo-

5    lated to a common altitude grid. These biases are then applied to the OMPS-LP time series in such a way to conventionally keep the SCIAMACHY mean level as absolute reference as follows:

$$OMPS_{deb}(\underset{\sim}{lat}\phi, \underline{lon}\theta, z) = OMPS(\underset{\sim}{lat}\phi, \underline{lon}\theta, z) - BIAS_{OMPS}(\underset{\sim}{lat}\phi, \underline{lon}\theta, z) + BIAS_{SCIA}(\underset{\sim}{lat}\phi, \underline{lon}\theta, z)$$    (2)

In this way, any offset between SCIAMACHY and OMPS-LP is accounted for with the help of MLS as a transfer standard. The merging is then achieved by concatenating the two data sets, in terms of ozone number density, and computing average

10    values from SCIAMACHY and OMPS-LP over the two months of overlap, i.e. February–March 2012. We exclude all bins

where the number of observations is lower than 10 or where the measurements from one of the instruments are not available. Figure 3 shows relative differences between the merged data set and MLS time series (after the subtraction of its bias with respect to SCIAMACHY) as a function of latitude for several altitudes.

Relative differences for the 'plain-debiased' merged time series are computed as follows:

$$Rel\,Diff(lat\phi, lon\theta, z) = (Merged(lat\phi, lon\theta, z) - MLS(lat\phi, lon\theta, z))/(Merged(lat\phi, lon\theta, z) + MLS(lat\phi, lon\theta, z)) * 200$$

$$(3)$$

[revised manuscript text omitted]